# A frog with three sex chromosomes that co-mingle together in nature: *Xenopus tropicalis* has a degenerate W and a Y that evolved from a Z chromosome

**Benjamin L. S. Furman**[1,2], **Caroline M. S. Cauret**[1], **Martin Knytl**[1,3], **Xue-Ying Song**[1], **Tharindu Premachandra**[1], **Caleb Ofori-Boateng**[4], **Danielle C. Jordan**[5], **Marko E. Horb**[5], **Ben J. Evans**[1]*

**1** Department of Biology, McMaster University, 1280 Main Street West, Hamilton, Ontario, L8S 4K1, Canada, **2** Department of Zoology, University of British Columbia, 6270 University Blvd Vancouver, British Columbia, V6T 1Z4 Canada, **3** Department of Cell Biology, Charles University, 7 Vinicna Street, Prague, 12843, Czech Republic, **4** CSIR-Forestry Research Institute of Ghana, Kumasi, Ghana, **5** Eugene Bell Center for Regenerative Biology and Tissue Engineering and National *Xenopus* Resource, Marine Biological Laboratory, 7 MBL St, Woods Hole, MA 02543 USA

* evansb@mcmaster.ca

**Data Availability Statement:** The RRGS and RNAseq data from *Xenopus tropicalis* have been deposited in the Short Read Archive of NCBI

## Abstract

In many species, sexual differentiation is a vital prelude to reproduction, and disruption of this process can have severe fitness effects, including sterility. It is thus interesting that genetic systems governing sexual differentiation vary among—and even within—species. To understand these systems more, we investigated a rare example of a frog with three sex chromosomes: the Western clawed frog, *Xenopus tropicalis*. We demonstrate that natural populations from the western and eastern edges of Ghana have a young Y chromosome, and that a male-determining factor on this Y chromosome is in a very similar genomic location as a previously known female-determining factor on the W chromosome. Nucleotide polymorphism of expressed transcripts suggests genetic degeneration on the W chromosome, emergence of a new Y chromosome from an ancestral Z chromosome, and natural co-mingling of the W, Z, and Y chromosomes in the same population. Compared to the rest of the genome, a small sex-associated portion of the sex chromosomes has a 50-fold enrichment of transcripts with male-biased expression during early gonadal differentiation. Additionally, *X. tropicalis* has sex-differences in the rates and genomic locations of recombination events during gametogenesis that are similar to at least two other *Xenopus* species, which suggests that sex differences in recombination are genus-wide. These findings are consistent with theoretical expectations associated with recombination suppression on sex chromosomes, demonstrate that several characteristics of old and established sex chromosomes (e.g., nucleotide divergence, sex biased expression) can arise well before sex chromosomes become cytogenetically distinguished, and show how these characteristics can have lingering consequences that are carried forward through sex chromosome turnovers.

(BioProject PRJNA627066) as has the RNAseq data from *Xenopus borealis* (BioProject PRJNA616217). Representative Sanger sequences have been deposited in GenBank (accession numbers MW115652-MW115842). The transcriptome assembly has been deposited at DDBJ/EMBL/GenBank under the accession GIVH00000000.

**Funding:** This work was supported by the Natural Science and Engineering Research Council of Canada (RGPIN-2017-05770) (BJE), Resource Allocation Competition awards from Compute Canada (BJE), the Whitman Center Fellowship Program at the Marine Biological Laboratory (BJE), the Museum of Comparative Zoology at Harvard University (BJE), and National Institutes of Health grants R01-HD084409 (MEH) and P40-OD010997 (MEH). The funders had no role in study design, data collection and analysis, decision to publish, or preparation of the manuscript.

**Competing interests:** The authors have declared that no competing interests exist.

## Author summary

Sex chromosomes often come in pairs (e.g., an X and a Y, or a Z and a W) and variation among species evidences widespread rapid evolutionary changes of sex chromosomes. To understand why, we examined a rare example of a frog (*Xenopus tropicalis*) with three sex chromosomes. We discovered a small sex-linked sliver of the genome that has a high proportion of genes with higher expression in males than females during gonadal differentiation. Molecular variation in expressed transcripts from this genomic region suggests that this pattern stems from decreased or lost expression of alleles on the W chromosome combined with a recent origin of the Y chromosome from an ancestral Z chromosome. These findings are consistent with theoretical expectations associated with reduced genetic recombination, and demonstrate that features of ancestral chromosomes have persistent genomic effects that bleed through sex chromosome transitions.

## Introduction

During eukaryotic evolution, genetic control of sexual differentiation changed many times [1]. In some instances, the establishment of a new master regulator for sexual differentiation is associated with cessation of recombination, and extensive divergence in nucleotides, gene content, and gene expression between non-recombining regions of each sex chromosome [2–7]. In other species, extensive recombination between sex chromosomes may occur, and gene content, function (in terms of gene expression), and cytological appearances of each sex chromosome may be almost identical (e.g., [8]). Between these extremes, there exists an astonishing range of variation in the extent of recombination suppression and the degree of sex chromosome divergence [6]. For those sex chromosome pairs that do diverge, it is unclear how fast and in what order differences between them arise. The ability to cope with differences between the sexes in the dosage of gene products stemming from degeneration of sex-linked alleles on the W or the Y chromosome [9], periodic recombination [10], and genomic conflict associated with mutations with sexually antagonistic fitness effects via the origin of sex-biased expression patterns [11] all may influence whether or not—and how much—sex chromosomes diverge from each other.

Another phenomenon that may influence recombination and divergence between sex chromosomes is turnover, wherein the genomic location, genetic function (i.e., whether female or male determining) or gene that triggers sexual differentiation changes [12, 13]. A sex chromosome turnover is considered "homologous" when a new variant that assumes the role of sex determination arises on an ancestral sex chromosome [14–17] and "non-homologous" if it establishes on a different chromosome pair from the ancestral sex chromosomes. Homologous and non-homologous turnovers may involve a new variant taking over with the opposite effects of an ancestral sex determining locus; this changes which sex is heterogametic (females for WZ systems, males for XY systems). For example, in medaka fishes, a new trigger for female development replaced an ancestral trigger for male development, creating a turnover of XY to WZ sex chromosomes [18]. Turnovers can also occur via translocation of a sex determination allele, which is the case in strawberries [19] and some salmonids [20]. Non-homologous XY to XY turnovers may be favoured by natural selection if the ancestral Y chromosome has a high load of deleterious mutations due to genetic degeneration [21, 22]. However, Y-linked deleterious mutations may disfavour an XY to WZ transitions if this results in homozygotes for the ancestral Y chromosome [16].

Understanding why sex determination systems and their associated sex chromosomes change is a challenging prospect (reviewed in [17]), but catching them in the act—during evolutionary windows where multiple sex determination systems co-exist in one species—may help us understand why and how this occurs. Specifically, these transition periods may offer insights into whether and how characteristics of ancestral sex chromosomes (e.g., nucleotide divergence, sex-biased expression, degeneracy) affect the evolution of the sex chromosome systems that follow.

In amphibians, many changes between male and female heterogamy have been inferred [15, 23–25], making this group a compelling focus for studies of new sex determining systems and early evolutionary events of young sex chromosomes. Within-species variation in the heterogametic sex has been identified in a handful of amphibians such as the Japanese wrinkled frog, (*Glandirana rugosa*; [14]), Hochstetter's frog (*Leiopelma hochstetteri*; [26]), and the Western clawed frog (*Xenopus tropicalis*; [27]), studied here. In *X. tropicalis*, W, Z, and Y chromosomes have been identified [27–29], but no cytological divergence among sex chromosomes of this or any other *Xenopus* species has been detected [30]. Most of the sex chromosomes of *X. tropicalis* are pseudoautosomal regions where genetic recombination occurs [31]. Current understanding is that the W is dominant for female differentiation over the Z, and the Y is dominant for male differentiation over the W [27]. WW and WZ individuals develop into females and WY, ZY, and ZZ individuals develop into males [27]. Thus females carry at least one W chromosome but not all males carry a Y chromosome. Although it is technically no longer a Z chromosome after the Y chromosome appeared, we nonetheless use this term following [27] as a placeholder to refer to the extant non-male-specific sex chromosome that descended from the ancestral Z chromosome. In principle, YY offspring could be generated if a genetic male (WY or ZY) was sex reversed and developed into a phenotypic female and then crossed with another genetic male. To our knowledge, natural sex reversal has not been reported in *X. tropicalis*, and we assume here that this is rare.

The genomic location of the female-associated region of the W chromosome was narrowed down using genetic mapping in a laboratory strain to a 95% Bayes credible interval positions 0–3.9 megabases (Mb) on chromosome 7 in genome assembly 9.1 (v9) [29]. However, this region was not completely linked to the female phenotype in that study, and it was proposed that this lack of complete linkage could stem from ancestral admixture with an individual carrying a Y chromosome [29]. The male determining factor of the Y chromosome of *X. tropicalis* is thought to be in a similar location as the female-determining factor [27], but its precise location has not been determined. Within the genus *Xenopus*, the most recent common ancestor of subgenus *Silurana*, which includes *X. tropicalis*, probably had heterogametic females [25]. This implies that the Y chromosome of *X. tropicalis* is younger than the W chromosome and Z chromosome, and thus derived from an ancestor of one of these chromosomes. Mitochondrial genomes of species in subgenus *Silurana* diverged about 25 million years ago [32], implying that the Y chromosome of *X. tropicalis* is younger than that. This information raises the possibility that *X. tropicalis* is currently in the midst of a homologous sex chromosome turnover.

*X. tropicalis* is a model organism for study of developmental biology and human disease [33–35]. Y chromosomes have been detected in laboratory strains of *X. tropicalis* that are thought to originate in Sierra Leone, Ivory Coast, Nigeria, and Cameroon [27], although this has not been confirmed directly in specimens sampled in nature. Also unknown is whether populations with male and female heterogamy geographically overlap and interact genetically in nature, whether the variation that defines these chromosomes occurs in the same gene or genomic region, or exactly when cytogenetically undifferentiated sex chromosomes, such as those of *X. tropicalis*, acquire characteristics that are often associated with old sex chromosomes (nucleotide divergence, sex-biased gene expression). Thus, the goals of this study are to

(i) test whether there is male or female heterogamy in natural populations of *X. tropicalis*, (ii) narrow down the region of sex linkage in this species, (iii) evaluate genome-wide patterns of sex-biased expression and nucleotide differentiation, and (iv) characterize patterns of recombination across the genome and between the sexes of wild-caught individuals of this species.

## Results

We used reduced representation genome sequencing (RRGS) to assess population structure and sex chromosome differentiation in wild-caught and georeferenced laboratory *X. tropicalis* individuals from Ghana, Sierra Leone and Nigeria. Then, to explore sex chromosome evolution and sex-linkage, we generated three families at McMaster University from imported wild caught frogs from Ghana, and their offspring. Nucleotide divergence, sex-linkage, and recombination was evaluated in two families using RRGS and Sanger sequencing, and nucleotide divergence and gene expression were evaluated in offspring from the third family using transcriptome sequencing (RNAseq). There are differences in the v9 genome assembly used by [29] and the v10 genome assembly used in this study in the sex-linked region of chromosome 7 (S1 Fig). To facilitate comparison to other studies, we report genomic coordinates of both assemblies for the $F_{ST}$ results and below for the sex-linkage results. Other findings discussed below from RRGS and RNAseq are reported using v10 coordinates and the genome-wide recombination analyses were performed using v9.

### Population structure in *X. tropicalis* and a small region of sex chromosome differentiation

We first tested whether there was a genome-wide signature of population differentiation in *X. tropicalis* samples derived from wild and georeferenced laboratory animals. This analysis identifies population differentiation between samples from Sierra Leone and Ghana + Nigeria with two partitions, and between Sierra Leone, Ghana west, and Ghana east + Nigeria with three (Fig 1). With more than three partitions, additional subdivisions are found within individuals from each geographic locality.

We then quantified $F_{ST}$ between females (n = 12) and males (n = 26) over a moving average of 50 SNPs in wild individuals from Ghana, and georeferenced lab individuals from Sierra Leone and Nigeria. Population structure coupled with different geographic sampling of males and females should have a genome-wide effect on $F_{ST}$ between females and males. There are two possible sex chromosome genotypes in females (WZ, WW) and three in males (ZZ, ZY, WY), and six possible parental genotype combinations (Fig 2). Therefore, in sex-linked regions, differences in allele frequencies and nucleotide divergence between the W, Y, and Z chromosomes are expected to cause $F_{ST}$ to be higher than elsewhere in the genome, including compared to the pseudoautosomal regions of the sex chromosomes.

Across the genome, the 95% CI for $F_{ST}$ is 0.002—0.038. Over the sex-linked region identified below and elsewhere [29], the mean $F_{ST}$ value is 0.049 (standard deviation = 0.023), which is significantly higher than the value over the entire genome. The highest $F_{ST}$ value in the genome (0.13) was present at position 9,940,000 in the sex-linked region of chromosome 7 in v10; $F_{ST}$ was >0.09 from positions 9,775,600–9,999,600 (Fig 3, S2 Fig). The locations of this $F_{ST}$ peak and range are 1,615,479 and 1,454,645–1,664,477 in v9, respectively. The $F_{ST}$ peak and the male-specific SNPs on the *X. tropicalis* Y chromosome discussed below (Table 1) overlap with a small genomic window of strongly female-linked variation on the *X. tropicalis* W chromosome found previously [29] (S1 Fig). Specifically, the margins of the $F_{ST}$ peak overlap with the most strongly female-linked genomic region (linkage group super_547:0; positions 1,365,917–1,693,249 in v9; LOD score: 13.13296453 [29]).

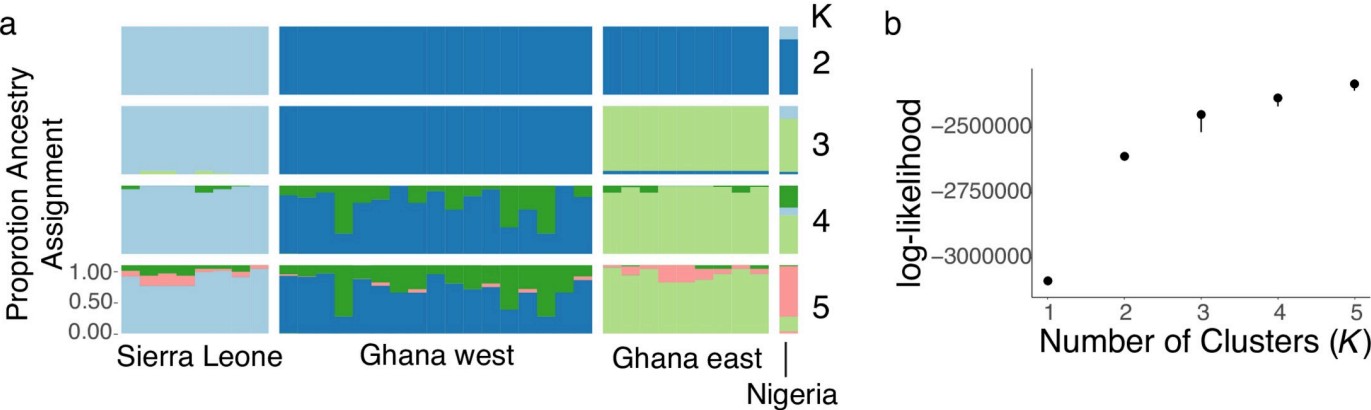

**Fig 1. Genetic cluster analysis of RRGS data illustrates geographic structure of wild *X. tropicalis*.** (a) Ancestry assignments of individual samples for 2–5 populations (K). (b) log-likelihood of values of K from 1–5.

## Evidence of genetic degeneration of the sex-linked region of *X. tropicalis*, and that the Y chromosome is derived from an ancestral Z chromosome

In *X. tropicalis*, one combination of parental sex chromosomes (WZ x ZY) produces a 1:3 female:male offspring sex ratio (WZ daughters and ZZ, ZY, or WY sons; Fig 2); this type of family does not have completely sex-linked genetic variation passed from either parent to all of the same-sex offspring (because the W and Z chromosomes are both inherited by sons and daughters, and the Y chromosome is not inherited by some sons). Another parental combination (WW x ZZ) produces only WZ daughters. This parental combination, and one other with no offspring sex-bias (WZ x ZZ), are expected to have completely female-linked genetic variation passed from mother to daughters on the W chromosome. The three other parental combinations (unshaded in Fig 2) are all expected to have no sex-bias in offspring numbers, and also

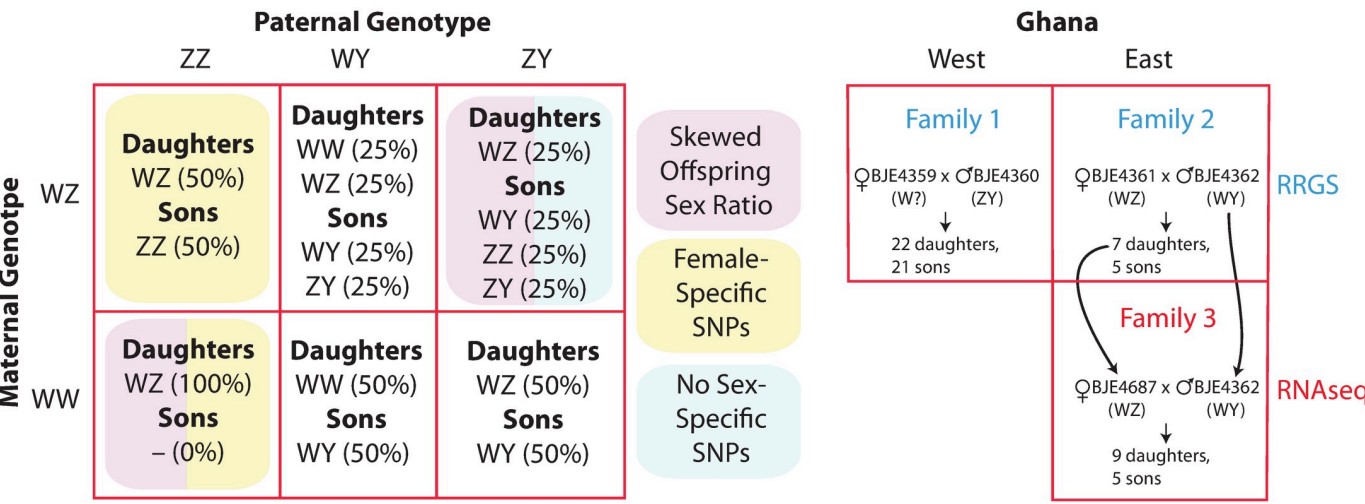

**Fig 2.** The three sex chromosomes of *X. tropicalis* can be crossed in six ways to produce offspring with different types of sex-linkage and/or skewed offspring sex ratios (left). Crosses on the left that are not shaded are expected to have male-specific SNPs passed from father to all sons in the male-specific portion of the Y chromosome. We generated three laboratory families from west and east Ghana for RRGS and RNAseq analyses (right). For the RNAseq analysis (Family 3), offspring were analyzed from a cross between the father and a daughter from Family 2 (indicated with arrows). On the right, putative sex chromosome genotypes described in main text are in parentheses with a question mark indicating either a W or a Z chromosome.

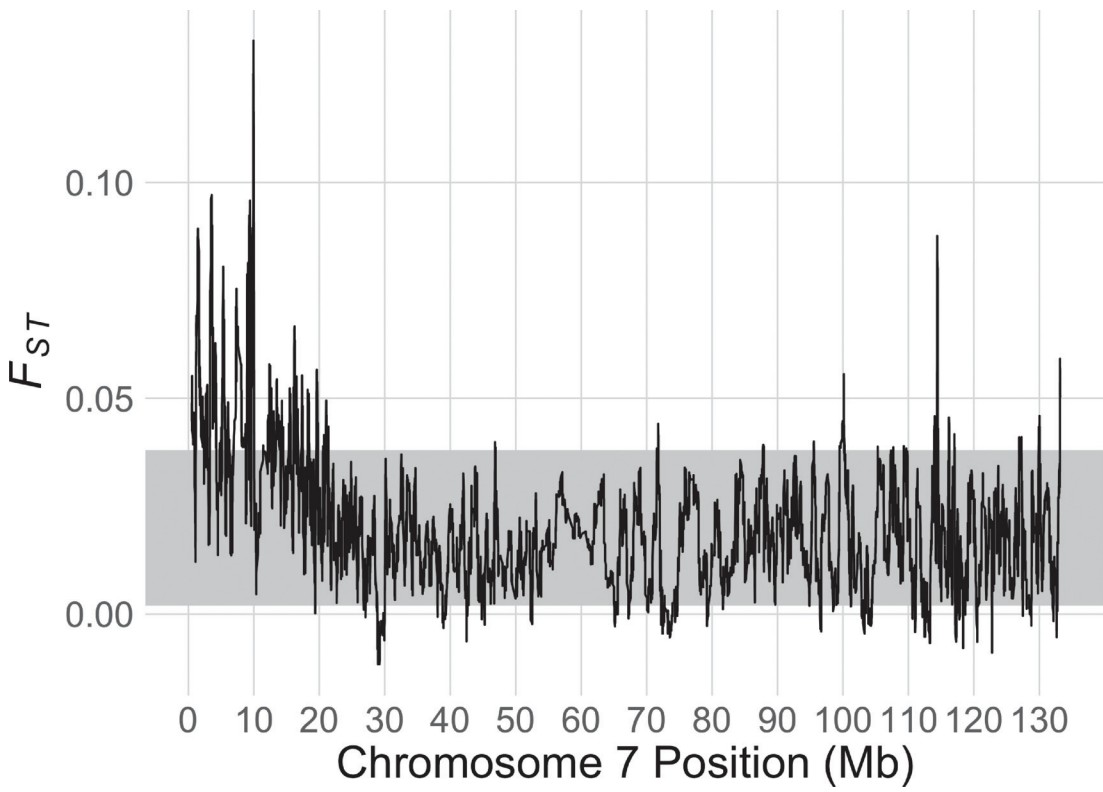

**Fig 3. $F_{ST}$ between females and males for *X. tropicalis* chromosome 7 of wild samples from Ghana, and georeferenced lab strains from Nigeria and Sierra Leone.** The grey band represents the whole genome bootstrap confidence intervals for the mean $F_{ST}$ that were generated by resampling $F_{ST}$ measured on the autosomes.

completely male-linked genetic variation passed from father to sons on the Y chromosome: WZ x WY, WW x ZY, WW x WY.

In the RRGS data from two families (Family 1 and 2) that were generated from wild caught individuals from the western and eastern edges of Ghana, genome-wide inheritance of single nucleotide polymorphisms (SNPs) provides unambiguous evidence for a sex determining system where males carry a Y chromosome. There were no maternal heterozygous sites that were sex linked prior to or after FDR correction in either family. However, five paternal sex-linked RRGS markers were found in the region between 8.1 Mb and 13.5 Mb on chromosome 7 of v10 in Family 1, and three paternal sex-linked RRGS markers were found in the region between 2.7 Mb and 6.54 Mb on chromosome 7 in Family 2. (Fig 4, S3 Fig). We intentionally sampled a subset of offspring with approximately equal numbers of each sex in Family 1 and 2 (22 daughters and 21 sons for Family 1, seven and five daughters and sons for Family 2). The presence of sons allows us to rule out the possibility that either of these two crosses was between a WW mother and a ZZ father. Together these observations demonstrate that the father of each of these families carried a Y chromosome, and that at least one of the parents in both crosses did not carry a Z chromosome, because any combination with both parents carrying one or more Z chromosomes would not have any completely male-linked SNPs (Fig 2). Thus the sex chromosome genotypes of Families 1 and 2 both could be any one of the three unshaded crosses in Fig 2. Additional details about polymorphism and sex-linkage in Family 1 and 2 are provided in S1 Text.

**Table 1. Results of the Sanger sequencing survey of 18 amplified regions (Locus) for seven groups of *X. tropicalis*: a lab cross and wild individuals from west Ghana (Family1, GWwild) or east Ghana (Family2, GEwild) that were used for the RRGS data (but not the RNAseq data), and captive strains at the National *Xenopus* Resource from Ivory Coast (IC) and Nigeria (Nigerian and Superman).** The genomic position of each locus in the *X. tropicalis* v9 and v10 are indicated, with the chromosome or scaffold followed by a range of genomic coordinates. For each group, the number of males and females sequenced are separated by slashes, followed by whether a male-specific SNP was detected (Y) or not (N); "NV" indicates no variation in the sequences. A dash indicates that the amplification was not attempted or that Sanger sequences were not clean. For two loci from the Family 2, an asterisk indicates that 4/5 males had a male-specific SNP and in both of these amplicons, the same male individual did not have this SNP; thus variants at these loci were almost but not completely sex-linked.

| Locus | v9 | v10 | Family1 | GWwild | Family2 | GEwild | IC | Nigerian | Superman | Notes |
|---|---|---|---|---|---|---|---|---|---|---|
| - | scaffold_486: 109006-109688 | Chr7: 590989-591647 | – | 3/2/N | – | 2/1/N | – | – | – | |
| - | scaffold_1093: 25288-26168 | Chr7: 2401099-2401979 | – | 3/2/N | – | 4/1/N | – | – | – | |
| vwa2 | scaffold_132: 209950-210334 | Chr7: 3988998-3989381 | – | 1/1/Y | – | 2/0/NV | – | – | – | a |
| bag3 | scaffold_83: 134640-135403 | Chr7: 6350671-6351443 | – | – | – | 2/1/N | – | – | – | |
| LOC108644867 | scaffold_130: 572284-572814 | Chr7: 7445611-7446140 | – | 3/2/NV | – | – | – | – | – | |
| phc1 | scaffold_130:760304-761055 | Chr7: 7992897-7992184 | – | 2/2/N | – | 1/1/NV | – | – | – | |
| LOC100488897 | scaffold_130: 643554-643884 | Chr7: 8109808-8110138 | 2/2/N | 5/5/N | 5/7/Y | 5/1/N | – | – | – | |
| aicda | Chr7: 665453-665920 | Chr7: 9026686-9027153 | – | 4/3/N | – | 3/1/N | – | – | – | |
| LOC116406517 | Chr7: 901880-902194 | Chr7: 9256905-9257219 | 2/2/NV | – | 5/7/N* | 1/1/Y | – | – | – | |
| LOC100127624 | Chr7: 1364981-1365454 | Chr7: 9677066-9677539 | – | 4/3/N | – | 3/1/N | – | – | – | |
| grp162 | Chr7: 1386997-1387487 | Chr7: 9698865-9699356 | – | 4/3/N | – | 3/1/N | – | – | – | |
| LOC116412229 | Chr7:1928340-1928761 | Chr7: 10256773-10257194 | 2/2/N | 4/5/N | 5/7/N* | 3/1/Y | 3/5/N | 5/7/Y | 10/7/Y | b |
| LOC116412144 | Chr7: 2045411-2045931 | Chr7: 10389186-10389706 | – | 4/3/N | – | 3/1/N | – | – | – | |
| prkg1 | Chr7: 5207535-5208256 | Chr7: 13469130-13469851 | – | – | 5/7/N | 1/1/Y | – | – | – | |

ᵃ: This is possibly an allele-specific amplification in some populations; no amplification occurred in several male and female individuals, and in the Ghana west population, one male sequence differed from one female sequence by 1 homozygous nucleotide; sequences from two Ghana east males were invariant and identical to the Ghana west male.

ᵇ: The same SNP was present in Superman and Nigerian males, but a different SNP was present in Ghana east males. The male-linked allele in Superman and Nigerian males amplified weakly, but consistently.

There were no informative RRGS markers between 6.54 Mb and 11 Mb on chromosome 7 of v10 in Family 2, so it was not possible to assess whether RGGS markers in this region were also sex-linked. However, genotyping of additional markers in Family 2 by Sanger sequencing found three completely or almost completely sex-linked markers located between 8.1 Mb and 10.26 Mb, suggesting that this region is sex-linked in both Families 1 and 2 (Table 1, Fig 4). By contrast, informative RRGS markers between 0 and 8.1 Mb on chromosome 7 of v10 were present in Family 1, but were not sex-linked in this family even though this region was sex-linked in Family 2 (Fig 4).

Analyses discussed below allowed us to conclude that the sex chromosome genotype of the father of Family 2 (BJE4362) was WY. Although we were not able to discern whether the sex chromosome genotype of the father of Family 1 (BJE4360) was WY or ZY, we suspect that his sex chromosome genotype was ZY, and that recombination occurs between the Z and Y chromosomes <8 Mb, but not at all or rarely between the sex-linked regions (<10.3 Mb; see below) of the W and Y chromosomes (and possibly not also between the sex-linked regions of the W and Z chromosomes, though we do not attempt to address this possibility here). This scenario would explain why there were sex-linked sites on the end of chromosome 7 in Family 2 but not Family 1. It is also consistent with evidence presented below for an origin of the Y chromosome from an ancestral Z chromosome (because recombination is more likely to occur in the sex-linked regions of closely related chromosomes), and also with degeneration of the sex-linked portion of the W chromosome (which is associated with recombination

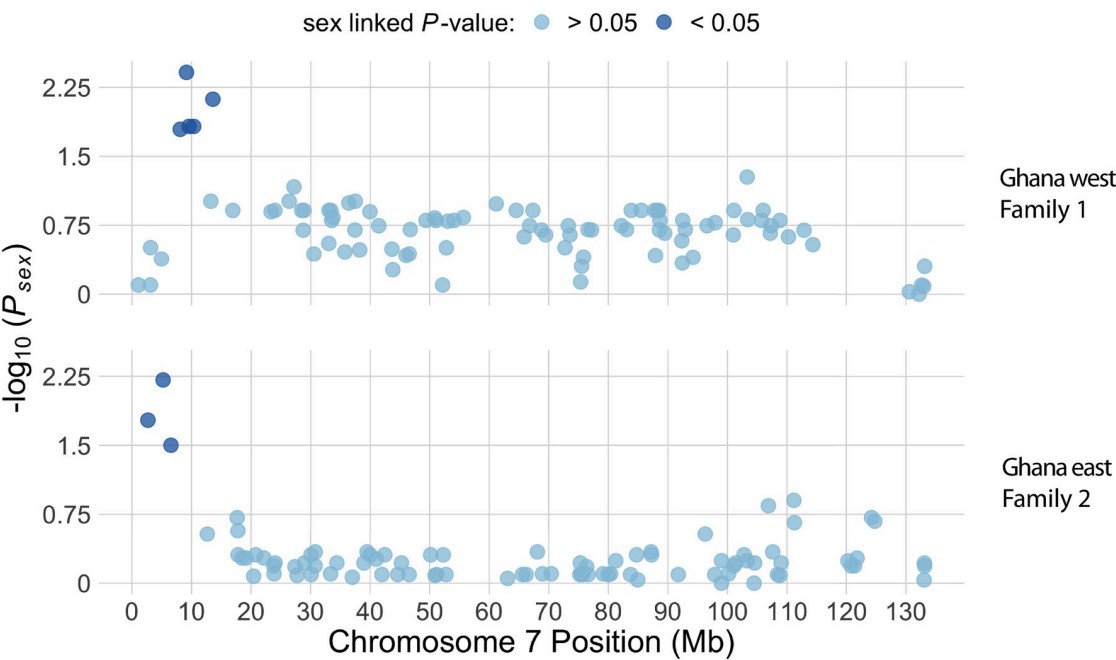

**Fig 4. Manhattan plot of association between genotype and sex phenotype for chromosome 7 in Family 1 from Ghana west (top) and Family 2 from Ghana east (bottom) for paternal heterozygous sites.** For both families, light dots indicate variants that are not significantly associated with sex, and dark dots indicate significant associations with sex after FDR correction (top) or before FDR correction (bottom). As discussed in the main text, we did not apply FDR correction for Family 2 due to a smaller dataset.

suppression) as a mechanism for a high density of transcripts with male-biased expression that is also discussed below.

Under this scenario of parental sex chromosome genotypes, recombination between the Z and Y chromosomes during spermatogenesis could cause some of the sons to not carry Y-linked SNPs at LOC100488897 (Table 1), which is located at ∼8Mb, even though they may have inherited the Y-linked male determining factor that is located between 8 and ∼10.3 Mb on chromosome 7; we do not precisely know the upper boundary of the sex-linked region, but 10.3 Mb is not sex-linked in wild individuals from Ghana east and west (Table 1), and most male-biased transcripts, discussed below, are encoded by genes <10.3 Mb (Table A in S1 Text). In Family 2, the lack of complete sex-linkage at two loci <10.3 Mb on chromosome 7 was due to homozygous genotypes in one son (four other sons had sex-linked SNPs in hetero-zygous genotypes). RNAseq data discussed below suggests that the mother of Family 2 was WZ, which would mean that two different sex chromosome genotypes are expected in sons of Family 2 (see top center of left side of Fig 2). One possibility is that the one unusual son was ZY and the other sons were WY, and that their heterozygous positions were due to divergence between the W and Y chromosomes that was not present in the ZY son.

One amplicon (LOC116412229) had a male-specific SNP in two strains from Nigeria (Nigerian and Superman) and a small sample of Ghana east wild individuals, but not in other strains we surveyed. In the strains from Nigeria, males were A/G and females G/G at position 1,928,777 or 10,257,211 in v9 or v10, respectively. A male-specific SNP is not definitive evi-dence of a Y chromosome because a segregating polymorphism on a Z chromosome or an autosome could by chance be present only in males. However, without invoking Y linkage, the chance of observing a heterozygous genotype in 15 of 15 males and none of 14 females is very

low (P $< 2 \times 10^{-9}$). The sex-linked SNPs in the Nigeria strain are in different genomic positions from the nearby sex-linked SNP in three wild Ghana east males. Overall, while the extent of the Sanger sequencing data were limited by difficulties with obtaining clean sequences from our amplicons, the findings from the available data are generally congruent with the results from the RRGS data in the sense that part of chromosome 7 <10.3 Mb appears to be partially or completely sex-linked in Family 2.

We also examined genotypes in sex-linked expressed transcripts of each individual offspring of a third family (Family 3) using RNAseq data in order to detect transcripts expressed from only one of the individual's sex chromosomes (based on observing no heterozygous variants), and those co-expressed by both alleles in heterozygous genotypes [17]. The RNAseq data was from tadpole stage 50 gonad/mesonephros. Although this study does not explore this issue, the initial motivation for selecting this tissue and developmental stage was that it corresponds with the timing of gonadal differentiation in *X. laevis* [36] and the sex determining gene of *X. tropicalis* could also be expressed in this tissue type and developmental stage. These tissues were dissected from tadpoles of Family 3 which was generated from a cross between the wild-caught father of Ghana east Family 2 that was used for the RRGS analysis (BJE4362), and a daughter of this cross (BJE4687; Fig 2). The sex of each tadpole was assessed using Sanger sequencing of amplicons in the sex-linked region (beginning at coordinates 8,109,808 and 9,256,905 in v10; Methods and Table 1). Although we did not assess sex-linkage in Family 3, evidence discussed above indicates that the father carried a Y chromosome. There was not a male-sex-bias in offspring numbers (we ended up sequencing transcriptomes from nine daughters and five sons) which, following the same reasoning above, indicates that at least one of these parents did not carry a Z chromosome, and that the sex chromosome genotypes of the parents of Family 3 could be any one of the same three unshaded crosses in Fig 2 that were possible for Families 1 and 2.

For two of the three possible sex chromosome genotype combinations (WW mother x WY father; WW mother x ZY father), if recombination is suppressed in the sex-linked portion of the W chromosome, we expected divergent sites in expressed transcripts of sex-linked genes to be similar within each offspring sex. This is because there is only one sex chromosome genotype for each sex for each of these parental sex chromosome genotypes (bottom middle and bottom right of the left side of Fig 2). However, for offspring from a WZ mother and a WY father, daughters and sons each have two possible sex chromosome genotypes (WW or WZ for daughters, WY or ZY for sons; top middle of the left side of Fig 2). If recombination is suppressed in the sex-linked portion of the W chromosome, we would expect that this type of cross could have two distinctive levels of within female nucleotide diversity in sex-linked expressed transcripts (in WW and WZ daughters), and also two distinctive levels of within male nucleotide diversity in sex-linked expressed transcripts (in ZY and WY sons). After filtering (Methods), we retained for analysis an average of 782 (range: 653–904) expressed transcripts from the sex-linked region (<10.3 Mb on chromosome 7 in v10) per individual, and an average of 50 bp (range: 43-57) per transcript within each individual.

Consistent with our predictions associated with a cross between a WZ mother and a WY father in Family 3, we observed two distinct levels of nucleotide diversity in expressed transcripts of sex-linked genes within daughters, and also two within sons (Fig 5). In addition to resolving the sex chromosome genotypes of the parents of Family 3 (the mother was WZ and the father was WY), these results indicate that the genotype of the mother of Family 2 (BJE4361, Fig 2) was also WZ because her daughter, who was the mother of Family 3 (BJE4687) carried a Z chromosome, and her father did not. These findings also indicate that the Y chromosome is derived from the Z chromosome and not from the W chromosome

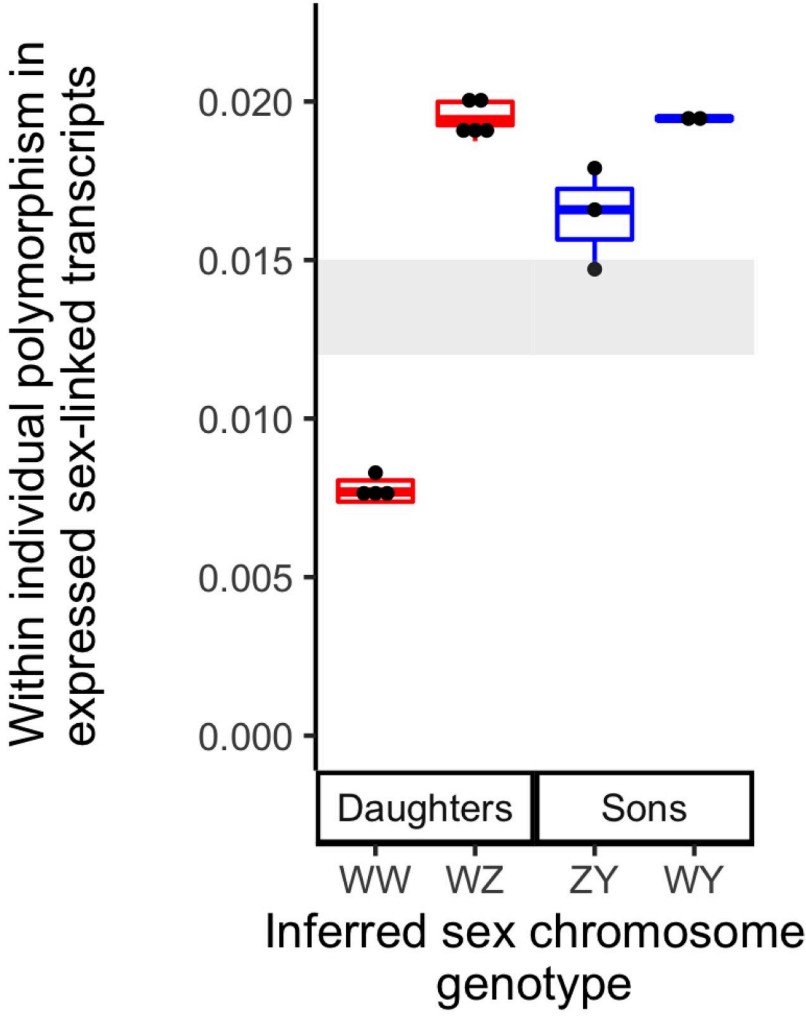

**Fig 5. In daughters and sons of Family 3, two distinct levels of within individual polymorphism in expressed sex-linked transcripts imply that there are two distinct sex chromosome genotypes in offspring of each sex.** Inferred sex chromosome genotypes (x-axis) are based on within individual polymorphism of expressed sex-linked transcripts (y-axis). The range of pairwise nucleotide diversity for non-sex-linked transcripts in the 14 individuals for which RNAseq was performed is depicted in gray.

because divergence between the Z and Y chromosomes is lower than divergence between the Z and W or between the Y and W chromosomes (Fig 5). Additionally, and perhaps most surprisingly, these results demonstrate that W, Z, and Y chromosomes all co-occur in nature in individuals from the same small seep (<6 feet wide) in east Ghana.

## The sex-linked portion of the *X. tropicalis* sex chromosomes has a very high density of genes with male-biased expression

Using the RNAseq data from Family 3, we then analyzed whole transcriptome expression from the gonad/mesonephros complex during an early stage of sexual differentiation. We used these expression data to evaluate how genes with sex-biased expression are distributed over the genome, including in sex-linked and non-sex-linked portions of the sex chromosomes. A total of 259,197 transcripts were assembled that mapped to one of the 10 chromosomes in v10; 296

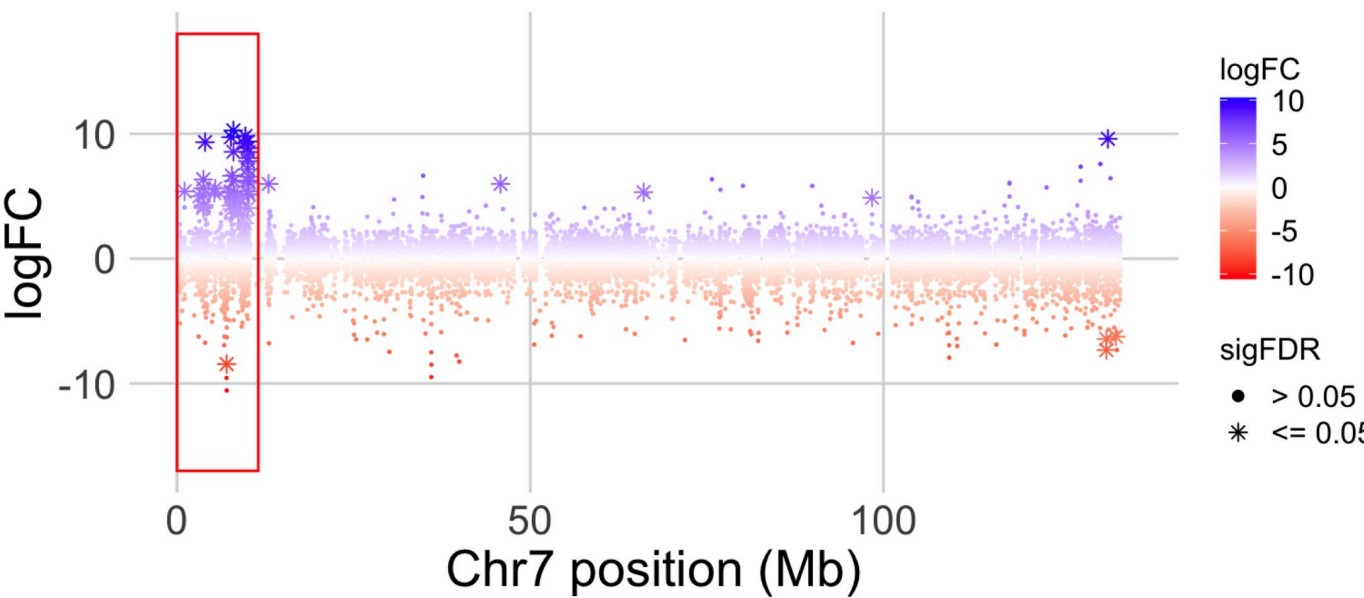

**Fig 6. Log₂ transformed male/female transcript expression ratio (logFC) along *X. tropicalis* chromosome 7 in offspring of Family 3.** The x-axis indicates the genomic coordinates in millions of base pairs (Mb). Small dots represent individual transcripts and * represent transcripts that are significantly differentially expressed after FDR correction (sigFDR). Positive values reflect male biased expression, negative values are female biased. A red box highlights a cluster of genes on the sex-linked portion of chromosome 7 with mostly male-biased expression.

transcripts were detected that mapped to scaffolds that were not assigned to chromosomes, and 2,816 did not map to these assemblies. Of these, 151, 1, and 5, respectively, had significant sex-biased expression after FDR correction.

In the non-sex-linked portion of the genome (including the pseudoautosomal region of chromosome 7), the numbers of transcripts with significantly male- or female-biased expression were relatively similar (n = 63 and 44, respectively). However, there were two genomic regions with a high density of genes with sex-biased gene expression (Fig 6, S5 Fig). The first is the sex-linked portion of chromosome 7 (<3.3 Mb in v9 or <10.3 Mb in v10), which has a very high density of genes with male-biased expression (45 transcripts from 30 genes in v10) but not female-biased expression (1 transcript; Table A in S1 Text, Fig 6). Twenty-seven of these transcripts from 20 genes were male-specific (expressed only in males); 18 transcripts from 13 genes were male-biased (expressed in both sexes but significantly higher in males), and only one transcript was female-specific. The proportions of differentially expressed transcripts from this region with male-biased or male-specific expression are significantly higher than expected based on the proportion from the rest of the genome (P < 0.00001, binomial tests). As a consequence of the high density of these genes on the sex-linked region, the number of transcripts with significantly male-biased or male-specific expression was far higher on chromosome 7 than any of the other chromosomes (Table B in S1 Text), even though the proportion of this chromosome that is sex-linked is small (<10% of chromosome 7; Fig 4; [31]), and even though chromosome 7 is intermediate in size. We also explored the effect of contrasting expression in subsets of male and female offspring based on inferred sex chromosome genotypes (S1 Text, S6, S7, S8 and S9 Figs). These analyses also suggest that some transcripts encoded by gametologs on the sex-linked portions of the W, Z, and Y chromosomes are differentially regulated, presumably due to a combination of divergence and polymorphism in the regulatory regions of genes in the sex-linked regions of these sex chromosomes.

The second region with a high density of sex-biased transcripts is on chromosome 3 between 114—128 Mb in v10; this area encodes a high density of female-biased transcripts (21 transcripts from 14 genes) but not male-biased (one transcript; Table C in S1 Text). However, the density of sex-biased transcripts on this region of chromosome 3 (1.5 transcripts/Mb) is substantially lower than the density of sex-biased transcripts on the sex-linked portion of chromosome 7 (4.4 transcripts/Mb) (S5 Fig). We do not know why this region has an atypically high density of female-biased transcripts.

## Why do genes in the sex-linked region encode so many transcripts with male-biased expression?

There are several possible explanations for the strong skew towards male-biased expression of transcripts encoded by genes in the sex-linked region of these frogs. One possibility is that this particular region had a high density of male-biased transcripts in an ancestor when this region was not sex-linked (that is, prior to the origin of a sex-determining locus on chromosome 7 in *X. tropicalis*). To gain perspective into this possibility, we turned to expression data that we collected for another study from *X. borealis*, a closely related allotetraploid species, from the same tissue (gonad/mesonephros) and a similar developmental stage (tadpole stage 48) [37]. We determined genomic locations of assembled transcripts in the *X. laevis* genome assembly version 9.2 using the same methods as described here for *X. tropicalis*, and as described in more detail elsewhere [37]. Because *X. borealis* is allotetraploid and because it has different sex chromosomes than *X. tropicalis* (on chromosome 8L [38]), this species has two autosomal chromosomes (chromosomes 7L and 7S), that are orthologous to the sex chromosomes of *X. tropicalis*. Inspection of genomic regions in *X. borealis* that are orthologous to the sex-linked region in *X. tropicalis* identified only one significantly male-biased transcript on *X. borealis* chromosome 7L, one on *X. borealis* chromosome 7S, and no significantly female-biased transcripts on either of these chromosomes (S10 Fig). This comparison does not rule out the possibility that a strongly male-biased expression skew was present ancestrally but lost during evolution of *X. borealis*, but it does suggest that there is no reason to expect that transcripts in this genomic region are somehow predisposed to have male-biased expression. Taken together, these comparisons favor the interpretation that the evolution of male-biased expression occurred in concert with the origin of sex-linkage <10.3 Mb on chromosome 7 in *X. tropicalis*.

## Rates and locations of recombination are sex-specific in *X. tropicalis*

We used RRGS data from Families 1 and 2 to compare genome-wide rates and locations of crossover events in females and males. For both families, the total length of the female linkage map greatly exceeded that of the male map, even though the female and male maps had a similar number of markers and spanned similar proportions of the genome. For the Ghana west family, the female map length was 920 cM (including 1504 SNPs), and the male map was 367 cM (including 1645 SNPs). For the Ghana east family, the female map length was 1495 cM (2061 SNPs), and the male map length was 630 cM (1857 SNPs). This indicates that recombination is far more common during oogenesis than during spermatogenesis.

There was a positive relationship between linkage map length and the physical size in base pairs (bp) of the genome assembly in female maps, but this was not evident in male maps. The slope of this relationship in females from Family 1 is 28.7 (95% confidence interval (CI): 6.7–50.7) and in females from Family 2 is 50.9 (CI: 28.5–71.8), whereas this slope for males from Family 1 is 7.5 (CI: -13.7–28.6) and males from Family 2 is 17.6 (CI: -3.9–39.0; Fig 7A). That male recombination rates were unrelated to the size in bp of the genomic region to which the linkage map corresponds, argues that their crossover events in males occur in more

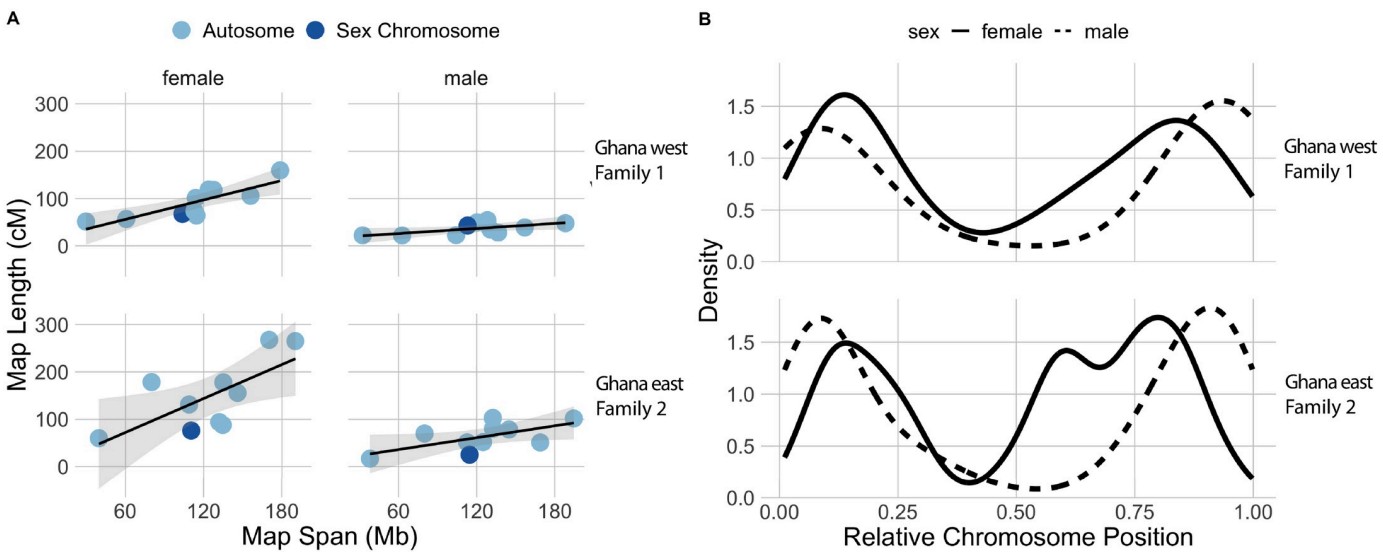

**Fig 7.** A) Linkage map length in centimorgans (cM) is positively correlated with the length of the genomic region in millions of base pairs (Mb) in females (left) but not in males (right) from Family 1 and Family 2 (top and bottom rows, respectively). B) Crossover density is more strongly biased towards chromosome tips in males than females in Family 1 and Family 2.

concentrated genomic regions, as compared to females. Consistent with this, crossover events were more biased towards chromosome tips in during male recombination compared to female recombination, although both sexes had a lower density of crossover events near the centers of chromosomes as compared to the first and third quartiles (Fig 7B). These differences are unlikely to be related to sex-differences in coverage of the RRGS data because markers included in the linkage maps spanned similar proportions of the chromosomes in both sexes, and in both populations (Family 1: an average of 95.1% of the chromosome lengths were covered for females and 93.2% for males; Family 2: 99.0% for females and 98.8% for males). We did not detect a substantial disparity in the number of crossovers on the sex chromosome (chromosome 7) in male maps of either family (Fig 7A), which is consistent with most of this chromosome being pseudoautosomal [31].

## Discussion

### The geographical context of *X. tropicalis* sex chromosomes

*Xenopus tropicalis* is distributed in tropical habitats in West Africa, ranging from Sierra Leone to western Cameroon [39]. Rainforest habitat in West Africa is interrupted by savanna in a region called the Dahomey Gap, which lies roughly in the center of the distribution of *X. tropicalis*, including southeastern Ghana, the southern portions of the countries of Togo and Benin, and southwestern Nigeria [40, 41]. Few records of *Xenopus* are available from the Dahomey Gap, and it is possible that there is a discontinuity in the range of *X. tropicalis* in this region. Limited data from mitochondrial DNA sequences suggests that there may be population subdivision within *X. tropicalis* that is associated with the Dahomey Gap [31]. Genome-wide data analyzed here point more strongly to subdivision between *X. tropicalis* populations from Sierra Leone and Ghana + Nigeria, and suggest populations from east Ghana, which were sampled in a forested patch within the Dahomey Gap, are less differentiated from populations from Nigeria (east of the Dahomey Gap) than from populations from west Ghana (west of the Dahomey Gap; Fig 1).

Our results identify, for the first time, a Y chromosome in *X. tropicalis* individuals sampled directly from nature (two localities in Ghana) and suggest a Y chromosome is present in a laboratory strain from Nigeria. In a recent study of sex-linkage in *X. tropicalis* that included the Nigeria laboratory strain that was used for the genome sequencing, the nature of the crosses did not permit assessment of whether a Y chromosome was present [29]. However, the authors concluded that if there was a Y chromosome, it would have originated from the Nigeria strain that was used in their cross [29]. Our limited Sanger sequencing survey of variation in a laboratory strain from Ivory Coast, which is situated between Sierra Leone and Ghana, did not identify sex-linked SNPs (Table 1). This does not rule out the possibility that the Ivory Coast strain also carries a Y chromosome because some of the males may have two Z chromosomes. Our findings from the RRGS and RNAseq data also provide the first georeferenced evidence of a Z chromosome in wild caught individuals (in west and east Ghana). Overall, these results demonstrate that the W, Z, and Y chromosomes co-mingle in the natural range of *X. tropicalis* in Ghana, and perhaps elsewhere. A key question motivated by these co-mingling sex chromosomes asks how and why they co-exist when they are associated with substantial offspring sex-ratio skew [27], which is expected to often be disadvantageous [42–45]. It may be the case that the Z chromosome has a low frequency in the populations we sampled in Ghana, and that the W chromosome segregates in these populations, more or less, like an X chromosome. Eventual extinction of the Z chromosome would transition the W chromosome into a new X chromosome, which is one way to prevent offspring sex-ratio skew with the new sex determining system associated with the newly emerged Y chromosome. Alternatively, if the Y chromosome were rare in a population, most crosses also would have a balanced sex ratio governed by the W and Z chromosomes. Extinction of the Y chromosome thus could also prevent offspring sex-ratio skew via reversion to the ancestral (WZ/ZZ) system for sex determination. Results presented here provide direct or indirect evidence that all three sex chromosomes were present in Families 1, 2, and 3, but does not quantify the frequencies of each of these chromosomes in natural populations. Further efforts to genotype sex chromosomes of *X. tropicalis* sampled in nature could evaluate these possibilities.

## Are the three sex chromosomes of *X. tropicalis* defined by different variants at one gene?

The three sex chromosomes of *X. tropicalis* could be part of an evolutionary transition, with a new system for sex determination on its way to fixation. Alternatively, this variation may be stable over evolutionary time, with different frequencies of each sex chromosome favored by unique factors in distinct environments [46]. In two laboratory bred families from Ghana and a laboratory strain from Nigeria, we identified male-specific SNPs that fall within the tightly female-linked region on the W chromosome of another *X. tropicalis* strain [29] and also a closely related species, *X. mellotropicalis* (S1 Text; [25]). That the male-specific region of the Y chromosome and the female-specific region of the W chromosome occur in gametologous locations (Fig 4) along with a peak of nucleotide differentiation (Fig 3, S1 Fig) points to the possibility that the female-specific (W-linked) and male-specific (Y-linked) variation that drives female and male differentiation, respectively, could be in the same locus or possibly segmental duplicates that are closely situated.

Mechanistically, genetic sex determination can be realized in many ways [1] which could include, for example, a dominant sex-specific allele (such as *Sry* in eutherian mammals), or dosage mechanisms that involve differences in copy number of a shared allele (such as *Dmrt1* in birds). At this point we can only speculate about mechanisms in *X. tropicalis*. One possibility is that there is a combination of these mechanisms, such as a loss of function allele for male

differentiation on the W chromosome (which causes WZ individuals to be phenotypically female and ZZ individuals to be male). However, this is inconsistent with previous findings that WZZ triploids develop into females [27]. Additionally, WWY triploids develop into males, which argues against sensitivity of the Y chromosome to dosage of the W [27]. A more plausible alternative is that the W chromosome carries female-determining allele whose function is not present on the Z chromosome, whereas the Y chromosome carries a dominant negative regulator of the female-determining allele on the W. A dominant negative regulatory role has been proposed for *dm-w*, which is a W-linked trigger for female differentiation in *X. laevis* over the male-related *dmrt1* gene (which is autosomal) and closely related to *dmw* by partial gene duplication [36, 47]. Future efforts aimed at identifying the variants that trigger female and male variation in *X. tropicalis* is crucial to unravel their fascinating evolutionary histories and genetic interactions.

## Signs of genetic degeneration in cytologically indistinguishable sex chromosomes

During gonadal differentiation, a total of 151 transcripts in the gonad/mesonephros transcriptome were identified with significant sex bias and a known genomic location; one third of these transcripts (n = 50) were in the sex-linked region of chromosome 7 in v10 (Fig 6, S5 Fig), which comprises <1% of the 1.7 Gb genome of *X. tropicalis* [48]. Of the transcripts in this genomic region with significantly sex-biased expression (n = 46), almost all were male-specific (n = 27) or male-biased (n = 18), none were female-biased, and only one was female-specific (Table A in S1 Text). An excess of sex-linked genes with male-biased expression was also observed in adult tissues, although that excess was not significant [49], which is a possible consequence of the lower quality genome assembly that was available at that time for determining the genomic locations of transcripts. Sex-biased expression of sex-linked transcripts in multiple developmental stages has also been observed in fish, and may be a more effective mechanism for resolving genomic conflict in broadly expressed transcripts than differential expression orchestrated by steroid hormones [50].

In the sex-linked portion of the *X. tropicalis* sex chromosomes there exists more substantial nucleotide divergence compared to the rest of the genome (Fig 3, S2 Fig) and divergence between expressed transcripts encoded by sex-linked genes on the W chromosome and the Z or Y chromosomes is higher than that between expressed transcripts encoded by sex-linked genes on the Z and Y chromosomes (Fig 5,). This information, combined with the observation that the closely related tetraploid species *X. mellotropicalis* has a female-linked genomic region in a homologous location to *X. tropicalis* [25], suggests that the Y chromosome evolved from the Z chromosome rather than from the W chromosome. If the Y chromosome eventually fixes in *X. tropicalis* (and the Z goes extinct), the mechanism of turnover would appear to follow the scenario depicted in Table 1D of [51] but with an ancestor with female heterogamy and a descendant with male heterogamy. In the absence of dosage compensation, sex chromosome turnover may be favored due to the accumulation of deleterious mutations and associated lowered or lost expression of alleles on the non-recombining sex chromosome [21, 22]. However, in *X. tropicalis* this scenario does not appear to apply since the degenerate W chromosome is staged to survive a transition to male heterogamy if the Y chromosome fixes in the future because, if this happened, the W chromosome would become an X chromosome.

Several factors have the potential to influence regulatory evolution on sex chromosomes, such as faster-X or faster-Z effects [52, 53]). The faster-X effect may be heightened in species with dosage compensation [52], although there is no strong evidence of dosage compensation in amphibians [54]. Evidence presented here is most consistent with degeneration of the W

chromosome, presumably prior to the origin of the Y chromosome from the Z chromosome, as a mechanism for male-biased expression of transcripts encoded by sex-linked loci. For example, a comparison between putative WW females and WY males identifies more substantial sex-biased expression than a comparison between WZ females and ZY males (S6 and S7 Figs). We also detected higher nucleotide polymorphism in expressed transcripts encoded by genes in the sex-linked region of putative WZ and WY individuals than in transcripts encoded by non-sex-linked (autosomal and pseudoautosomal) transcripts. This is also suggestive of divergence due to recombination suppression on the sex-linked portion of the W chromosome. One prediction that is associated with the mechanism behind male-biased expression of sex-linked transcripts in *X. tropicalis* is that *X. mellotropicalis* should also have a degenerate W chromosome and also exhibit male-biased expression in the sex-linked portion of its sex chromosomes. This is another interesting direction for further exploration.

## Sex differences in recombination

In *X. tropicalis* from Ghana, the rate of recombination is higher during oogenesis than spermatogenesis, and the crossover densities vary during these meiotic events as well, with proportionately more crossovers occurring in more central region of chromosomes during oogenesis compared to spermatogenesis (Fig 7B). This pattern was evident in a relatively small sample of crossover events that were observed in two biological replicates, but are congruent with results recovered from the other two *Xenopus* species examined so far—*X. laevis* and *X. borealis* [55]. A lower density of crossover events in the center of chromosomes was also detected in another study of *X. tropicalis* [29], although sex-differences in these densities were not evaluated in that study. Overall, this suggests that these sex-biases in recombination rate and location are widespread in *Xenopus*, including across ploidy levels (*X. laevis* and *X. borealis* are both allotetraploid), and probably as well the most recent common ancestor of extant *Xenopus*. In several other species, including other frogs, the recombination rate is also higher in females compared to males, though the opposite pattern has also been observed [56–62]. Paternal crossovers are more concentrated at the ends of chromosomes than maternal crossovers in other vertebrates as well, including humans [56–58, 61, 62]. Why sex differences in the locations of recombination exist is not entirely clear, but is mechanistically achieved by sex-differences in the rate that double strand breaks occur and in the rate that they are resolved into crossover or gene conversion events [62], which are influenced by the unique ways that meiosis occurs in females and males [63].

These sex-differences in recombination rate and location have interesting ramifications for the genomic positions of male-specific and female-specific variation on sex chromosomes. In females, triggers for female-determination should frequently be located on the ends of a W chromosome because there they should be disrupted less frequently by recombination as compared to alleles that are not near chromosome ends [62]. This prediction is supported by the W chromosome of *X. tropicalis* and in *X. laevis* where another female-determining gene – *dm-w* – is also positioned on the end of a chromosome (2L) [36], where the rate of recombination in females is relatively low [55]. The position of the male-determining factor on the end of the Y chromosome is not expected because recombination is higher in this region. However, it appears that suppressed recombination between the W and Z chromosomes <10.3 Mb was already in place prior to the origin of the Y, and this would presumably prevent disruption of the trigger for male differentiation on the Y.

## Outlook

We report here the co-occurrence of W, Z, and Y chromosomes in natural populations of *X. tropicalis* from Ghana. We identified a high density of transcripts with a strong skew towards

male-biased expression that originate from a small, differentiated, sex-linked genomic region in this frog. The findings of this study are consistent with the expectation that recombination suppression can lead to degeneration of sex chromosomes [64]. These results also evidence W chromosome genetic degeneration in a species with cytologically undifferentiated sex chromosomes, show a small male-linked region on the Y chromosome overlaps with a female-linked region of the W chromosome [29], and demonstrate that these three sex chromosomes co-occur in the same populations in nature. These findings open the possibility that variation at a single locus or a set of tightly linked loci define the three sex chromosomes of *X. tropicalis*, with alternative pairings of these variants governing whether an individual develops into a female or male. Exactly what genetic variation governs sex determination in *X. tropicalis* and how this variation is distributed across the natural range of this species remain uncharacterized, and are a promising direction for future efforts. Together, these features illustrate that several characteristics that are frequently attributed to old sex chromosomes (regulatory degeneration, nucleotide divergence) can in fact be present before divergence is detectable at the cytogenetic level, and persist through the evolutionary windows during which new sex chromosomes arise and replace ancestral sex chromosomes.

## Methods

### Genetic samples; reduced representation genome sequencing

To study sex-linkage, recombination, and population structure in *X. tropicalis*, we performed reduced representation genome sequencing (RRGS, [65]) on laboratory generated and wild caught individuals. The RRGS samples included 22 female and 21 male offspring from Family 1 whose parents were both from west Ghana (mother: BJE4359; father: BJE4360), seven female and five male offspring from Family 2 whose parents originated from east Ghana (mother: BJE4361; father: BJE4362), both parents from both of these families, 18 and seven additional wild caught samples from Ghana west and Ghana east, eight samples from individuals derived from Sierra Leone, and one from an individual derived from Nigeria. Parents of the lab crosses were performed at higher (∼four times) coverage than the offspring in order to increase the genotype quality in these individuals. Libraries were constructed with the *Sbf1* restriction enzyme (Floragenex, Portland, OR, USA), and multiplexed on one lane of an Illumina 2500 machine.

The wild *X. tropicalis* samples were collected from two locations near the western and eastern borders of Ghana: Ankasa Nature Reserve (GPS: 5.24424 -2.64044, altitude: 48 m; Ghana west), and near the town of Admedzofe (GPS: 6.83165 0.43642, altitude: 738 m; Ghana east). Offspring of animals from Ghana west and east are available upon request from McMaster University. Families from each population were generated by injecting parents with Human Chorionic Gonadrotropin (Biovendor, Asheville, NC, USA) to induce ovulation and clasping, and offspring were reared until post-metamorphic maturation. The Sierra Leone individuals (four females, four males) and a Nigeria individual (a female) were derived from georeferenced populations that were maintained at the Station de Zoologie Expérimentale at the University of Geneva [66]. The phenotypic sex of lab offspring were determined by surgical examination of gonads after euthanasia via transdermal overdose of MS222 (Sigma-Aldrich, St. Louis, MO, USA). The sexes of individuals from Ghana, Nigeria, and Sierra Leone were determined based on external morphology (females with larger size and larger cloacal lobes; males with nuptial pads on the forearms and smaller cloacal lobes). DNA was extracted from webbing, liver, muscle, or blood using the DNeasy blood and tissue extraction kit (Qiagen, Toronto, Canada).

RRGS reads from each individual were de-multiplexed using using Radtools [67], and trimmed with Trimmomatic version 0.39 [68], enforcing a minimum length of 36 bp,

removing 3 bp from the leading and trailing ends, and requiring less than four ambiguous bp in a sliding window of 15 bp. This resulted in an average of 5,000,000 reads per individual (range ~700,000–20,000,000). We aligned these data to the *X. tropicalis* genomes v9.1 and v10.0 using BWA [69], and used samtools/bcftools [70, 71] to call genotypes. Individual genotypes that did not have a minimum depth of 15 or had a genotype quality below 20 were set to missing. Additionally, all individual genotypes were discarded from a genomic position if >20% of laboratory offspring had missing genotypes, Hardy-Weinberg equilibrium in the lab offspring was violated, or >10% of lab offspring had a genotype that was not possible given the parental genotypes. This last category of sites are often a consequence of genotyping errors where heterozygous positions are called as homozygous [55, 72, 73]. For Family 1, we also filtered any individuals that had greater than 20% missing data, leaving 36 offspring; we did not apply the same quality filter to Family 2, because of the substantially smaller family size. Finally, we filtered the data to one randomly selected SNP per restriction-site associated region (RADTag) in each family.

The analysis of RRGS data involved mapping reads to a reference genome that was generated from a female individual of unknown sex chromosome genotype. Possible concerns with and justification of this approach are discussed in further detail in S1 Text.

### Analysis of sex-linkage

With the filtered SNP datasets for the two families aligned to v9 and v10, we calculated allelic association with sex following [74]; results were essentially the same for both genomes and v10 is presented here. This analysis was performed within each of our two families (Family 1 and 2) for bi-allelic sites that were heterozygous in the father or mother. Genotyping errors can reduce power to detect sex linkage, so we developed an approach to detect putative genotyping errors that resembled double recombination events in a small genomic window, and set them to missing data, thereby reducing their impact (additional details are provided in S1 Text). This substantially reduced the frequency of false positive signals of genotype association with sex (comparing S3 to S11 Figs).

In an attempt to narrow down the sex-linked region in populations of *X. tropicalis* with male heterogamy beyond the signal that was present in the RRGS data, we used Sanger sequencing to survey for sex-linked variants (Table 1). We analyzed both of our laboratory crosses, our wild caught samples from both of these localities in Ghana, and male and female individuals from a colony at the National *Xenopus* Resource, Woods Hole, MA, USA, that are thought to be derived from Ivory Coast (RRID:NXR_1009) and Nigeria (RRID:NXR_1018). We focused these efforts on genomic regions in the vicinity of the sex-linked regions that were identified by our RRGS analysis and [29, 31].

### Sex chromosome differentiation and population subdivision

For the samples originating from Sierra Leone, Nigeria, and Ghana—including the parents of each laboratory cross but not including the offspring of these crosses—we assessed admixture proportions by analyzing the RRGS data using NGSadmix version 32 [75]. We removed reads with a map quality <20 from the bam files, set the SNP_pval (likelihood of there being a SNP) parameter of NGSadmix to 1e-6, and used a minimum minor allele frequency (minMAF) of 0.05. We estimated genetic ancestry for partitions of 1–5 clusters (*K*), and ran 20 replicates for each value of *K*. We used CLUMPP [76] to combine the replicates while averaging the population assignments and correcting for label switching.

To test for differentiation of the sex chromosomes, we quantified $F_{ST}$ between females and males for each SNP following the bi-allelic $F_{ST}$ approach of [77]. Because we do not know the

sex chromosome genotype of almost all individuals for which we performed RRGS (both parents of Family 2 and the father of Family 1 are exceptions, see Results), we were unable to evaluate $F_{ST}$ between cohorts of females and males that each had the same sex genotype. Instead, we evaluated $F_{ST}$ between males and females across all samples for which we collected RRGS data, except the offspring of the two laboratory crosses. This included wild individuals from Ghana east (1 female, 8 males), Ghana west (6 females, 14 males), and georeferenced lab individuals from Sierra Leone (4 females, 4 males) and Nigeria (1 female).

## Transcriptome analysis

We dissected gonad/mesonephros tissue from 14 tadpoles at developmental stage 50 [78] that were offspring of Family 3, which had a wild caught father from Ghana east that was used in the RRGS (BJE4362) and one of his daughters from Family 2 (BJE4687). Tadpole stage 50 was chosen for analysis because this is the stage where expression of the sex determining gene *dm-w* has been detected in *X. laevis* [36]. The tadpole gonad/mesonephros tissue was preserved in RNAlater, and RNA was extracted individually from each sample using the RNeasy micro kit (Qiagen, Toronto, Canada). For each tadpole, we also preserved tail tissue in ethanol, and genomic DNA was extracted using the DNeasy kit.

Based on our results from RRGS and Sanger sequencing (see Results), the sex of each tadpole from the second Ghana east cross was determined based on the presence (males) or absence (females) of heterozygous genotypes in two completely or almost completely sex-linked amplicons (LOC100488897, primers: Scaf2_f1 + Scaf2_r2 and LOC116406517, primers: trop_east_SNP1_F1 + trop_east_SNP1_R1; Table 1; Table D in S1 Text). In Family 2, which was used for the RRGS data, the first of these amplicons had a sex-specific heterozygous SNP in five of five sons and none of seven daughters, and the second of these amplicons had an almost sex-specific SNP in four of five sons and none of seven daughters (Table 1). For the tadpoles that were used for RNAseq, heterozygosity at both of these amplicons was concordant for all individuals in the sense that heterozygosity was observed either at both amplicons or at neither amplicon (results from these tadpoles are not presented in Table 1 because we were not able to infer sex from adult individuals). This effort indicated that nine of the 14 tadpoles used in the RNAseq analysis were female and five were male. The accuracy of this indirect approach to sexing these tadpoles is evidenced by the very strong signature of expression divergence in the sex-linked region (Fig 6).

Library preparation and transcriptome sequencing was performed at the Centre for Applied Genomics (Toronto, Canada), multiplexing all 14 samples on one lane of an Illumina 2500 machine and 150 bp reads. Reads were trimmed using Trimmomatic version 0.36, removing the first and last three bases, retaining reads with a minimum length of 36 bp, and a 'maxinfo' setting of 30 and 0.7 (which determines the nature of an adaptive quality trim that aims to balance the benefits of preserving longer reads against the costs of retaining sequences that have errors). A *de novo* transcriptome assembled using Trinity version 2.8.2 with a minimum k-mer coverage of two.

Transcript counts were quantified for each sample using Kallisto v.0.43.0 following the methods of [79] with default parameters for indexing (using a kmer size of 31) and quantification (using quant parameter settings: -b 0 -t 1). We discarded genes with an average of less than one raw read per sample. Read counts from Kallisto were then used for differential expression between males and females with the EdgeR package version 3.4, using the vanilla pipeline (i.e., calcNormFactors, estimateCommonDisp, estimateTagwiseDisp, exactTest for comparison between males and females), following the EdgeR vignette [80]. EdgeR was used to calculate the $\log_2$-transformed male/female expression ratio (logFC), wherein values above

or below zero indicate genes that are more highly expressed in males or females, respectively. To avoid ratios equal to zero or undefined, a default prior count of 0.125 was added to all samples using the exactTest function of EdgeR. Using estimated read counts from Kallisto, we also performed an independent differential expression analysis with the DESeq2 package [81], following the DESeq2 vignette. Shrinkage was used with adaptive t prior shrinkage estimator from the package "apeglm" [82]. This option reduces the mean squared error of expression levels of each gene relative to the classical estimator, especially for genes with low expression levels [81, 82]. Because the results of the EdgeR and DeSeq2 analyses were similar (S12 Fig), we report only the EdgeR results.

We defined significantly sex-biased genes based on a false discovery rate (FDR) with Benjamini-Hochberg correction cutoff of 0.05 from the EdgeR output, and requiring the absolute value of logFC to be > 2. Genomic locations of individual transcripts were then ascertained based on the best match of each transcript against the *X. tropicalis* v10 (NCBI BioProject AAMC00000000.4; GenBank Assembly submission GCA_000004195.4) using a splice-aware aligner GMAP [83]. Median expression values for each sex were quantified after transcripts per million normalization (TPM) [84]. Confidence intervals for these medians were obtained using the DescTools package [85].

To quantify variation in expressed transcripts encoded by sex-linked genes, for each offspring from Family 3 we mapped the RNAseq data to the transcriptome assembly and called genotypes using bwa and bcftools. We filtered genotypes with < 4X coverage, genotype quality < 20, and map quality < 20. Pairwise nucleotide diversity was then calculated for each individual using a perl script and collated with transcript location as assessed above using R.

## Linkage mapping

In order to evaluate whether and how the rates and genomic locations of recombination differ between the sexes, we used the RRGS data to build and compare sex-specific linkage maps for each chromosome of the *X. tropicalis* families using Onemap v1.0 [86] and v9.1 [29]. Using the same approach as [55], we first identified the largest linkage group per chromosome using all genotypes (maternal-specific, paternal-specific, or both parents heterozygous), setting the minimum logarithm of the odds (LOD) score to five and the maximum recombination fraction to 0.4. We then separated heterozygous markers that were maternal-specific or paternal-specific for each chromosomal linkage map, and reconstructed sex-specific linkage maps for each chromosome, using a minimum LOD score of three. In this way we were able to reconstruct sex-specific rates and locations of recombination during oogenesis and spermatogenesis, respectively. Ordering of markers used in the linkage map was based on their mapping positions to the v9.1 genome.

After an initial build, we inspected individuals and set as missing data any single markers or sets of markers within a 10 Mb window that indicated a double recombination event, under the assumption that these genotypes are most likely due to genotyping errors because two recombination events are usually rare in very small genomic windows. We then reconstructed a sex- and chromosome-specific linkage maps with these filtered sets of markers. To determine how chromosome lengths (as covered by markers used in the linkage map) related to inferred map lengths for both families, we used a linear model with fixed effects of sex in which recombination occurs, family used for the linkage map, and interaction between those fixed effects, and a three-way interaction between sex, family, and amount of base-pairs covered by the extreme markers used in the linkage map (i.e.,

$map\ length \sim sex * family + sex : family : bp_{covered}$). Residuals were evaluated for non-

normality to ensure proper model fit. Analyses were performed in R using the `lm` function and confidence intervals were generated with `confint` [87].

### Ethics statement

This work was approved by the Animal Care Committee at McMaster University (AUP# 17-12-43).

### Supporting information

**S1 Data. Numerical data for figures.**
(ZIP)

**S1 Text. Supplemental methods, results, and Tables A–D.**
(PDF)

**S1 Fig. A dot plot of the sex-linked portion of chromosome 7 in v9 (y-axis) and the corresponding region in v10 (x-axis).** The most strongly female-linked linkage group identified by [29] (pink) the $F_{ST}$ peak identified here (dotted line; Fig 3). The 95% CI of the female-linked region from [29] has an inversion between assembly v9 and v10 at the upper bound of the most strongly linked region (super_547:1), and an insertion in v10. However, the most strongly sex-linked linked regions identified by [29] and this study are syntenous between these assemblies.
(EPS)

**S2 Fig. $F_{ST}$ between females and males for all *X. tropicalis* chromosomes provides perspective on the level of differentiation of the sex linked region of chromosome 7.** $F_{ST}$ was calculated and plotted as described in Fig 3.
(EPS)

**S3 Fig. Genome-wide sex linkage Manhattan plot for genotype association with sex for paternal heterozygous sites with correction for double recombinants (S1 Text).** For the Ghana west population (left), the *p*-values are FDR corrected, and for the Ghana east population (right), the *p*-values are not corrected (due to a much smaller sample size, see Methods).
(EPS)

**S4 Fig. Pairwise nucleotide diversity in expressed transcripts encoded by genes in the sex-linked (SL) region in individuals with different putative sex chromosome genotypes.** For comparison, data from the entire sex linked region from Fig 5 are displayed in blue next to values from the first (<6Mb) and second (6-11 Mb) portions of the sex-linked region; other labeling follows Fig 5.
(EPS)

**S5 Fig. The degree of sex-biased expression of gonad/mesonephros tissue in stage 50 tadpoles, expressed as the $\log_2$ transformed ratio of the male/female fold change (logFC) on each of the ten *X. tropicalis* chromosomes (labeled on the right) in offspring of Family 3.** The x-axis indicates the genomic coordinates of the transcript start position in millions of bp (Mb) on v10. Small dots represent individual transcripts and * represent transcripts that are significantly differentially expressed after FDR correction. Boxes indicate a cluster of genes on the sex-linked portion of chromosome 7 with mostly male-biased expression, and another cluster of genes on a portion of chromosome 3 with mostly female-biased expression.
(EPS)

**S6 Fig. Analysis of differential expression between putative WW females and WY males (subset 1) on chromosome 7.** Labeling follows Fig 6. The high density of genes encoding transcripts with sex-biased expression extends slightly beyond the region with a high density of male-biased transcripts that was identified in the RNAseq analysis of all samples (red box). In this analysis, the sex-linked region of chromosome 7 had 34 significantly male-biased transcripts with 27 of these being male-specific and 7 being expressed in both sexes; 3 transcripts in the sex-linked region were significantly female biased and all three were expressed in both sexes.
(EPS)

**S7 Fig. Analysis of differential expression between putative WZ females and ZY males (subset 2) on chromosome 7.** Labeling follows Fig 6. In this analysis, the sex-linked region of chromosome 7 had 19 significantly male-biased transcripts with 16 of these being male-specific and 3 being expressed in both sexes; there were no significantly female-biased transcripts detected in the sex-linked region.
(EPS)

**S8 Fig. Analysis of differential expression between putative WW females and ZY males (subset 3) on chromosome 7.** Labeling follows Fig 6. The high density of genes encoding transcripts with sex-biased expression extends slightly beyond the region with a high density of male-biased transcripts that was identified in the RNAseq analysis of all samples (red box). In this analysis, the sex-linked region of chromosome 7 had 37 significantly male-biased transcripts with 29 of these being male-specific and 8 being expressed in both sexes; 2 transcripts in the sex-linked region were significantly female biased and both were expressed in both sexes.
(EPS)

**S9 Fig. Analysis of differential expression between putative WZ females and WY males (subset 4) on chromosome 7.** Labeling follows Fig 6. In this analysis, the sex-linked region of chromosome 7 had 19 significantly male-biased transcripts with 16 of these being male-specific and 3 being expressed in both sexes; 1 transcript in the sex-linked region were significantly female biased and it was expressed in both sexes.
(EPS)

**S10 Fig. In the allotetraploid species *X. borealis*, genomic regions that are orthologous to the sex-linked region of *X. tropicalis* (boxes) do not encode transcripts with substantially skewed male-biased expression.** Data are from gonad/mesonephros tissue from *X. borealis* tadpole stage 48; labeling follows Fig 6. Assembly and expression analysis of these *X. borealis* data followed the same steps as for *X. tropicalis*, with the exception that the transcripts were mapped to the *X. laevis* genome assembly version 9.2 because a high quality assembly is currently unavailable for *X. borealis*. Orthology was established using dot plots as in S1 Fig, but using chromosome sequences from *X. tropicalis* and *X. laevis* instead of different genome assemblies of *X. tropicalis*. A comprehensive analysis of these *X. borealis* data is presented elsewhere [37].
(EPS)

**S11 Fig. Genome-wide sex linkage Manhattan plot for genotype association with sex for paternal heterozygous sites, without correction of double recombinants (S1 Text).** FDR correction and non-correction follows S3 Fig.
(EPS)

**S12 Fig. Differential expression analysis with EdgeR and DeSeq2 produced similar results as illustrated here for chromosome 7.** In the sex-linked region <10.3 Mb, both methods identified 32 male-biased transcripts, EdgeR but not DeSeq2 identified 13 additional male-biased and 1 female-biased transcripts, DeSeq2 but not EdgeR identified 2 additional female-biased transcripts, and neither methods identified significant sex-biased expression in 1,737 other transcripts on chromosome. 7.
(EPS)

## Acknowledgments

We thank Brian Golding for access to computational resources, and Jessen Bredeson, Sofia Medina Ruiz, Dan Rohksar and their colleagues, and Xenbase [88], for making the v10 *X. tropicalis* genome assembly available. We also thank the Associate Editor and three annonymous reviewers for constructive feedback on earlier versions of this manuscript.

## Author Contributions

**Conceptualization:** Benjamin L. S. Furman, Ben J. Evans.

**Data curation:** Benjamin L. S. Furman, Caroline M. S. Cauret, Martin Knytl, Xue-Ying Song, Tharindu Premachandra, Danielle C. Jordan, Marko E. Horb, Ben J. Evans.

**Formal analysis:** Benjamin L. S. Furman, Caroline M. S. Cauret, Martin Knytl, Xue-Ying Song, Tharindu Premachandra, Danielle C. Jordan, Marko E. Horb, Ben J. Evans.

**Funding acquisition:** Marko E. Horb, Ben J. Evans.

**Investigation:** Benjamin L. S. Furman, Caroline M. S. Cauret, Martin Knytl, Xue-Ying Song, Tharindu Premachandra, Caleb Ofori-Boateng, Danielle C. Jordan, Marko E. Horb, Ben J. Evans.

**Methodology:** Benjamin L. S. Furman, Xue-Ying Song, Ben J. Evans.

**Project administration:** Ben J. Evans.

**Resources:** Marko E. Horb, Ben J. Evans.

**Supervision:** Marko E. Horb, Ben J. Evans.

**Visualization:** Benjamin L. S. Furman, Caroline M. S. Cauret, Xue-Ying Song, Tharindu Premachandra, Ben J. Evans.

**Writing – original draft:** Benjamin L. S. Furman, Ben J. Evans.

**Writing – review & editing:** Benjamin L. S. Furman, Caroline M. S. Cauret, Martin Knytl, Xue-Ying Song, Tharindu Premachandra, Caleb Ofori-Boateng, Danielle C. Jordan, Marko E. Horb, Ben J. Evans.

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
