## [Decision Letter · Decision Letter 0]

7 Jun 2020

Dear Dr Evans,

Thank you very much for submitting your Research Article entitled 'High intensity sex chromosome evolution sexualized the transcriptome of a frog (Xenopus tropicalis)' to PLOS Genetics. Your manuscript was fully evaluated at the editorial level and by independent peer reviewers. The reviewers appreciated the attention to an important problem, but raised some substantial concerns about the current manuscript. Based on the reviews, we will not be able to accept this version of the manuscript, but we would be willing to review again a much-revised version. We cannot, of course, promise publication at that time.

If you decide to revise the manuscript for further consideration at PLOS Genetics, please aim to resubmit within the next 60 days, unless it will take extra time to address the concerns of the reviewers, in which case we would appreciate an expected resubmission date by email to plosgenetics@plos.org.

[LINK]

We are sorry that we cannot be more positive about your manuscript at this stage. Please do not hesitate to contact us if you have any concerns or questions.

Yours sincerely,

Catherine L. Peichel

Guest Editor

PLOS Genetics

Kirsten Bomblies

Section Editor: Evolution

PLOS Genetics

Comments from Guest Editor:

Your manuscript has been carefully reviewed by two experts in the field, as well as by me. We all agree that these polymorphic X. tropicalis sex chromosomes present a fascinating system for the study of sex chromosome evolution and that there are some really interesting results in this paper. Nonetheless, the paper needs substantial rewriting to clarify points of confusion about the data and to tone down some of the claims about what can be inferred from these data. Furthermore, some additional analyses (detailed below in my comments and those of the reviewers) would also greatly improve the clarity and impact of the manuscript.

1. Title: “High intensity sex chromosome evolution sexualized the transcriptome of a frog (Xenopus tropicalis)”

The title needs to be changed to better reflect the findings of the paper. I agree with Reviewer 2 that it is unclear what is meant by “high intensity sex chromosome evolution”. And, there is no evidence that the transcriptome of this frog has been sexualized. Indeed, there are transcripts linked to the sex determination locus that show differences in expression between males and females, but these are a very small proportion (~0.05%) of the transcriptome. Furthermore, we do not know if these genes showed sex-biased expression before the evolution of these sex chromosomes or whether populations with and without the Y chromosome discovered here differ in the extent of sex-biased expression of these transcripts. Without these data, it is impossible to assign causality.

2. L26-28: “These observations argue for a strong role for natural selection in sexualizing the transcriptome, with mutations in the sex-linked genomic region with sex-specific fitness effects being frequently and efficiently favored”.

Again, this is an overstatement of the findings. See comment 1 about the sexualization of the transcriptome. And, the transcripts that were identified in this study as sex-biased during gonadal differentiation have not been linked to sex-specific fitness effects.

3. L39-41: “Using modern genomic approaches, we discovered rapid and intense natural selection in a small sliver of the genome that sexualized the transcriptome during gonadal differentiation.”

There are no tests for selection presented in this manuscript. Divergence between sex chromosomes can also result from neutral processes, and disentangling the roles of neutral and selective processes on sex chromosomes requires careful analyses.

4. L41-43: “Our results point to the possibility that three genetic variants at a single gene define the different sex chromosomes of this frog”

This is certainly a very interesting possibility, but the data presented in this paper are far from conclusive and much more work would be required before this could be determined. I think it is perfectly fine to speculate on this possibility in the Discussion but would leave such far- reaching conclusions out of the Abstract and Author Summary and instead highlight the results that you have actually found.

5. L43-44: “These findings match theoretical preductions that many mutations have sex-specific effects on fitness”

Again, no sex-specific effects on fitness have been shown here. And it is possible that sex-specific expression of genes linked to the sex-determination locus could be due to other processes, including the possibility of different artefacts raised by the reviewers.

6. L59-61: “the extent of recombination suppression and the degree of sex chromosome divergence …are not necessarily positively correlated”

Certainly age and extent of recombination suppression/degree of sex chromosome divergence are not necessarily positively correlated, but I am unaware of any studies that show either a high degree of sex chromosome divergence in the absence of recombination suppression, or a low degree of sex chromosome divergence in the presence of recombination suppression. Of course, I am aware of the work in frogs in which sex chromosomes are not highly diverged, depsite suppression of recombination in males. But, rare recombination has been detected between these sex chromosomes, which is enough to account for the low levels of divergence. But, perhaps I have misunderstood the point or missed some key references. I tried to look at the review cited by the authors for this point, but could not find a clear statement in that review like the one in this manuscript.

7. L67-68: “wherein the function (i.e. whether female or male determining)”

This is confusing as stated here, because changes in heterogamety are discussed later in the paragraph, and because transitions can involve male-determining to male-determining factors (or female-determining to female-determining). I think here it might be more clear to say “wherein the identity or the genomic location of the sex determining locus changes”.

8. L76: perhaps also cite papers in salmonids for translocation of an existing sex determination gene; e.g. Yano et al. (2013) Evolutionary Applications 6: 486-496.

9. L131-133: I wasn’t sure why the YY individuals were mentioned here until reading further. I agree with Reviewer 1 that it might be nice to make a figure with the possible crosses and sexes of the different chromosome combinations. I drew one for myself!

10. L129-143: Within this paragraph alone, the different sex chromosome genotypes are referred to as ZW or WZ, YZ or ZY, and YW and WY! It is already challenging for the reader to follow these different combinations; please use one order for each genotype combination and then apply it consistently throughout the paper.

11. L144: Like Reviewer 2, I also stumbled over the point that three crosses are introduced earlier but only two are mentioned here. Perhaps in the very helpful overall introduction to the data (lines 120-127), this could be clarified.

12. L161-162: the statement: “and that at least one (and possibly both) of the parents did not carry a Z chromosome” needs more explanation. I am guessing that it is because the numbers are too small in the crosses to be confident that the sex ratios are really 1:1?

13. L174-197: I find this section confusing as well as troubling that the region of sex-linked markers differ between the two families. I appreciate that you did extensive Sanger sequencing to try and identify additional, shared markers that were sex-linked across the populations and in the two crosses. This was a lot of work and did not yield very conclusive results. It is not necessary to do so for this paper, but one option that has worked very well for my lab is to determine whether any of the SNPs identified in your RAD-seq data are in restriction sites. You can then design PCR primers flanking these SNPs and simply do a restriction digest of the PCR product and run the products on an agarose gel to diagnose whether an individual is homozygous for the presence of the allele that creates a cut site, heterozygous, or homozygous for the allele that does not contain the restriction site. Alternatively, you could scan the genomic sequence from this region for microsatellites, and design primers flanking those microsatellites, to see if any show patterns of sex-linkage. This approach has also worked well in my lab for identifying recombination boundaries on sex chromosomes.

For the paper, I think it could be more clearly explained that the Ghana East family shows that there is no recombination with SNPs between 2.6 and 6.5 Mb and sex phenotype: the next marker is somewhere after 10Mb, suggesting that the male-determining factor is within the first 10Mb of the chromosome; you just don’t have any markers to say where the boundary is. The marker most strongly associated in the Ghana West family suggests that the male determination factor is between 8.1 and 13.79, with the peak at 9.149. So, these intervals do overlap, but the data are shaky. One idea would be to look for sex-linked SNPs in the RNA-seq data of the third cross, which also was derived from the same Ghana East father as the mapping cross and see if additional resolution of the sex-linked region can be found. This would hopefully increase confidence that the sex-determination locus is indeed the same in these two populations/crosses.

14. L109-110: I agree with the Reviewers that these tests would likely be much more powerful if split between the different populations!

15. L208-220: This paragraph is very confusing. I think it would be much clearer for the reader if you first explain that there are differences in the order of the assembly v9 and v10 in this region of the genome, and that you are trying to compare the locations of markers identified as sex-linked in a ZW system with those identified here.

16. L237: “although we did not assess sex-linkage in this cross”. You certainly could do this with the RNA-seq data, using a method like SEX-Detector (Muyle et al. (2016) Genome Biology and Evolution 8: 2530-2543), which would provide additional markers and increase confidence in the concordance of the position of the sex determination locus in the two different populations from Ghana.

17. L241-244: This sentence is not particularly clear. Also you found 151 of 259,197 mapped transcripts were sex-biased (~0.05% of transcripts). The clustering of these transcripts is definitely interesting, but I do not see that this is the “sexualized transriptome” referred to in the title and abstract.

18. L248-249: “suggestive of stronger sexual selection in males”. This is an overstatement. Is a difference of 62 male-biased vs 43 female-biased transcripts of 259,197 total transcipts significant? And, what does expression during gonadal development have to do with sexual selection?

19. L267-286: I find it over-reaching to immediately jump to the sexual antagonism explanation for this result. First, you need to consider the possibility that there are artefacts that could explain this result. The possibility of mapping errors due to using a female genome without the Y chromosome absolutely needs to be considered (suggested by Reviewer 1) as does the possibility that the ancestral W is degenerated in this region and therefore the reduced expression in females simply results from missing sequence from the W chromosome (suggested by Reviewer 2). It is not clear whether the genome assembly was generated from a ZW or WW female; if a ZW female, it is possible that the assembly represents mostly the Z chromosome if the W has experienced degeneration. You might be able to assess this by examining your RAD-seq data: if all of the males in your crosses carry a degenerate W chromosome, then you would find reduced number of reads (relative to the genome-wide average) in your RAD-seq data in the sex-linked regions.

If the results are not due to an artefact, then why would metabolic genes like mannose-6-phosphate receptor (accounting for 3 of the sex-biased transcripts) or gapdh (accounting for 8 of the sex-biased transcripts) be under sexually antagonistic selection during gonadal development? Furthermore, I don't really understand the argument about why it is useful to compare the expression level of the sex-biased genes to the expression level of non sex-linked, non sex-biased genes? This argument needs to be clarified.

20. L307-309: It would be really interesting to present more detailed data on the male and female meiotic events in these two families in this region of chromosome 7. A supplementary table or figure would be fine. Is there any evidence for suppression of recombination in male meiosis in this region of the genome?

21. L346-348: If your hypothesis is correct that there are three different variants at the sex determination locus, then you would expect it to be present on all the sex chromosomes. In this case, it would be interesting to know whether any of the genes that are more highly expressed in developing male gonads are interesting candidate genes for the sex-determination locus? With the RNA-seq data, you should be able to look for sex-specific/sex-linked SNPs in these genes that might lead you to a good candidate gene!

22. L401-402: “pronounced nucleotide divergence”. I am not sure that an Fst of 0.05 counts as “pronounced”.

23. L465-467: Why is this surprising? If sexually antagonistic selection is driving these patterns, as you claim, the pseudoautosomal region is predicted to be an excellent place for sexually antagonistic mutations (See Otto et al. (2011) Trends in Genetics 27: 358-367; Jordan and Charlesworth (2012) Evolution 66: 505-512; Charlesworth et al. (2014) Evolution 68: 1339-1350).

24. L468-483: Here you should probably cite a recent review from Sardell and Kirkpatrick (2020) American Naturalist 195: 361-379, which provides a comprehensive overview of the generality of these sex-specific recombination patterns and possible explanations. Another nice and interesting review is Brandvain and Coop (2012) Genetics 190: 709-723.

25. L514-515 “It is fascinating that these pronounced sex-specific fitness differences can emerge so early in development – at or before the earliest steps in primary gonadal differentiation.”

Again, please be clear about what you have shown, which is sex-biased gene expression, not sex-specific fitness differences.

I know this is a lot of comments. I hope these suggestions are useful for you; they are certainly intended to be constructive and help you improve your manuscript!

Reviewer's Responses to Questions

**Comments to the Authors:**

Reviewer #1: I found this manuscript very hard to follow, and below I comment on some text that is too detailed to be understandable, or too long-winded. Line 116 states the main goals as studying X. tropicalis to

(i) test for male or female heterogamy in natural populations

(ii) narrow down the sex-linked region

(iii) study sex-biased expression [and nucleotide differentiation, which probably belongs under point ii]

(iv) characterize patterns of recombination in both sexes of wild-caught individuals of this species

The study seems to have detected a region that appears to suggest male heterogamety, whereas some previous work suggested female heterogamety. It does identify a sex-linked region, though it was not clear enough whether this is the same region in both these systems (the abstract suggests this, but the reasoning is not clearly explained). I have some reservations about the conclusions under aims (iii) and (iv).

It is no longer surprising or astonishing (lines 18 and 60) that genetic sex-determining systems differ (it’s not clear what ‘extensively’ means) among – and even within – species. This has been known for many decades, with many well-studied examples, and line 90 cites three other cases in frogs, — the question is whether the frog system studied can add new understanding. Line 83 says, correctly, that ‘Understanding the drivers of [or, better, the selective forces causing] sex chromosome turnovers is challenging (reviewed in [18]), but catching them in the act – during evolutionary windows where multiple sex determination systems co-exist in one species – may help us understand why and how they occur’. Yes, studies should be done, but please tell us here precisely what the questions are.

It is also confusing to write ‘three sex chromosomes’. Even the term’ sex chromosomes’ (and Y chromosome) may be misleading in the system studied, as these are sex-determining genes with none of the distinctive characteristics of sex chromosomes. If these terms are used, it needs to be explained that they are used purely for brevity, but that the situation is very different from that of more familiar sex chromosomes whose Y chromosomes include non-recombining regions.

The most interesting finding, which IS novel, is that a small ‘sex-associated’ part of the sex chromosome pair in this frog can be detected by higher sequence ‘differentiation’, but is not completely sex-linked, yet the region has a 50-fold enrichment of transcripts with male-biased expression during early gonadal differentiation, compared to the rest of the genome. The information about sex-differences in the rates and genomic locations of recombination is also valuable.

DETAILED COMMENTS

In line 198, the heading has a typo, and should read ‘The X. tropicalis Sex Chromosome Has a Small Region of Differentiation Between the Sexes’. Fst between the sexes can identify the sex-linked region much more precisely than genetic mapping in families (with limited progeny numbers, see comment below), and can also tell one which sex is heterogametic for the region. The ms states that all of the females have a W-chromosome, but does not explain how this is known, and whether it is consistent with the genetic results from Ghana frogs. [‘differences in nucleotide polymorphism …. Is’ should be corrected to be in the plural, though the sentence can be omitted entirely, and indeed most of the paragraph]. Something called v10 is mentioned, but not explained.

The paragraph starting in line 208 tells us that a peak in Fst occurs at the genome assembly location where a W factor was previously inferred. This text could be made shorter and clearer by simply referring to Figure 2 (and S2), together with mentioning which population it refers to, and the paper(s) with the previous genetic result. I was left unclear whether populations with male and female heterogamety all have an Fst peak in the same region, as Figure 2 seems to combine several population samples. It is also not clear what genome-wide Fst values are seen between the sexes in the sampled individuals. To understand the gene expression results, readers need to understand the extent of the region that has high Fst between the Y and X, or W and Z, depending on which system is being analysed. Figure 3 appears to show that at least 20 genes within a few Mb have high M/F ratios. It would be helpful to tell us the actual size of the region, and the raw fold difference represented by the y axis, so that we can know if these are all large differences (rather than just the log values, where differences are harder to see). It should be made clear whether this is a ‘normal’ gene density for this genome (or for a chromosome end), or out of line in either direction, and whether the GC content could be affecting the expression results (I do not know if this is a potential problem, but it is known to affect DNA sequence coverage estimates).

The gene expression results should also be evaluated carefully, and any other caveats should be discussed. A recent study found that sex differences were due to ‘reference bias’ in a species where the reference genome was a female in an XY species, and new work (by the same lab) that used reference genomes from the males used to estimate expression did not detect the expression differences that had been claimed previously (ZHOU AND BACHTROG 2012; WEI AND BACHTROG 2019). This possibility should be excluded in future work, including this study (where, of course, it may not be a concern, but, if so, readers should have it explained why not). I therefore did not review the section about expression in detail. If this concern is not justified, it should be explained why it can be excluded. In addition, the use of the gonad/mesonephros complex during an early stage of sexual differentiation should be justified. Clearly, such tissue is relevant to finding the sex-determining gene(s) but it is not clear to what extent other genes expressed only in gonad tissue will be included.

There seems to be no heading for the section about recombination (line 296 onwards), and the text does not actually describe mapping results, or the marker numbers, or really explain the reasoning for any conclusion. I was not sure whether linkage groups were based on enough markers to be confident that markers near the ends were included. The text says ‘There was a strong relationship between map length and chromosomal coverage in females map’ [meaning the female map of an unspecified family], but here ‘coverage’ seems to mean something that is not defined, and Fig. 4, which is stated to show the result, doesn’t use it. I think that the authors are trying to explain that total genetic map lengths in females increased with the chromosomes’ lengths [measured in an unspecified way, maybe the assembly lengths?], but in males were unrelated to these lengths, which suggests that crossover events in males are concentrated in limited regions. However, it is unclear whether the genetic and physical — assembly — maps are good enough to make this convincing.

Given these caveats, and the need for extensive shortenings and clarifications, I did not review the Discussion section.

MINOR COMMENTS [Please note that the line numbers are exceedingly small, and sometimes I may not have the correct one, as I could not always read them. The text is also small and very difficult to read on a laptop]

A general comment is that the text is long-winded and could be shorter and clearer. For example ‘A remarkably high number of changes amphibian sex chromosome systems is evidenced by variation among species in whether males or females are the heterogametic sex’. Means simply ‘In amphibia, many cases of changes between male and female heterogamety have been inferred [or discovered]’. Line 100 onwards could also be shortened and made clearer. The important point is that the ancestral sex determining system of subgenus Silurana, which includes X. tropicalis, was inferred to have female heterogamy, implying that the X. tropicalis Y is derived either from the W-chromosome or the sex chromosome that is shared with other species, which is referred to as the Z-chromosome [28].

Line 51: expropriated does not seem the right word

Line 61: It is confusing to say that differences in recombination suppression do not correlate with the

degree of sex chromosome divergence, as divergence will largely occur only once recombination is suppressed. It is also not clear enough to say that ‘dosage imbalances’ are a problem, as the main problem when Y or W chromosomes lose genes is that the amount of gene product in the affected sex may be lower than optimal, which is not what readers will understand from this phrase (they will think of the cases where balance is needed between the expression of two or more genes in a multi-protein complex, which may contribute to the evolution of sex chromosome dosage compensation, but are very rare). These problems can be corrected by shortening the Introduction to focus on the points relevant to this study, including the very long-winded description of turnover events, which can be briefly outlined, with a reference to a recent review.

Line 81: Is heterogamy the correct word?

Line 92: ‘segregating male9linked and female-linked variation’ is not correct terminology. It should be something like ‘variants that segregate in families as Y- or W-linked’, and ‘no cytological divergence among sex chromosomes of this or any other Xenopus species has been detected’, means ‘no Xenopus species with genetic sex-determination has heteromorphic sex chromosomes’.

Line 130: ‘dominant for female differentiation over the Z-‘ is not well expressed. Maybe change to something like ‘Relative to the Z, defined above, the factor carried by the ‘W’ dominantly determines femaleness and the ‘Y” factor is male-determining’. The following text is hard to understand and can be cleaned up and shortened (it repeats things unnecessarily). Maybe a Supplementary figure can show the expected progeny ratios of the crosses, and be referred to to state the conclusion clearly and briefly, for example, as follows (some comments are in []) ‘Two small families generated from individuals from the western and eastern edges of Ghana, provided unambiguous evidence for male heterogamety. The sex ratio was close to 1:1 in both crosses (22 females and 21 males or Ghana west, and seven females and five males for Ghana east). Several paternal SNPs at the distal end of chromosome 7 were associated with sex, and several SNPs were completely Y-linked in each family (though the sex-linked SNPs differed in the two families), but no heterozygous SNPs in the maternal frogs were identified as sex linked in either family (Fig. 1). Thus, at least one (and possibly both) of the parents in both crosses did not carry a Z-chromosome [here, I was not sure why this is inferred — should it be W chromosome?]. After FDR correction, five SNPs in the Ghana west sire were significantly associated with offspring sex [I am not sure why genome-wide ‘heterozygosity’, which is not defined, is mentioned here, as the significance test is surely all that is needed], and three in the Ghana east family. [It is important to understand that complete sex linkage in such small families does not mean that no recombination occurs in the region studied — to detect complete sex-linkage, one needs to use an adequate natural population sample.] Some non-sex-linked SNPs in the Ghana west family also varied in the Ghana east family, and were sex linked, and vice-versa, probably reflecting the small family sizes, such that recombination events may not occur in a given family; overall, the results identify a sex-linked region, and not a non-recombining region.

In small samples of wild caught individuals, Sanger sequencing identified at least one fully sex-linked SNP in the Ghana east Ghana east population, but it was not fully sex-linked in the Ghana west one, or in either family (Table 1). [Again, the text can be greatly shortened and much more clearly written].

References

Wei, K. H.-C., and D. Bachtrog, 2019 Ancestral male recombination in Drosophila albomicans produced geographically restricted neo-Y chromosome haplotypes varying in age and onset of decay. PLoS Genetics 15: e1008502. doi: 10.1371/journal.pgen.1008502

Zhou, Q., and D. Bachtrog, 2012 Chromosome-wide gene silencing initiates Y degeneration in Drosophila. Current Biology 22: 522–525. doi: 10.1016/j.cub.2012.01.057

Reviewer #2: X. tropicalis is a great system for studying sex chromosome turnover, as it has polymorphic ZW and "WY" sex-determining systems (derived from an ancestral ZW). In this manuscript, Furman et al use a combination of genetic and sequencing approaches to characterize the sex-linked regions of ZW and YW individuals. I really enjoyed the manuscript, but I struggled to interpret some of the data, to a large extent because not much information is provided/available on the ancestral pair of sex chromosomes. Specifically:

- An important part of the puzzle is how differentiated the ancestral ZW pair is, and whether the Y chromosome is derived from the Z or the W. I think this could really change the interpretation of the differentiation and gene expression patterns. For instance, if the W-specific region has reduced expression due to degeneration, and the Y is derived from the non-degenerated Z, then WW females may have lower expression than WY(Z) males. The fact that a similar excess of male-biased genes was found in ZW X. tropicalis (ref. #52) is consistent with this scenario.

- More information is in places needed to make sense of the results. For instance, it would be very helpful to state early on what is known about the sex chromosomes of the lab strains used here, and if at all inferred, those of the Sierra Leone and Nigeria individuals. If this is not known, then it should be explicitly mentioned as an important caveat.

- FST analysis: I found it confusing that all genotypes were combined; which sex chromosome is differentiated, if we don't know the genotypes of the individuals used for the Fst analysis? Would it be possible to plot separately the differentiation between:

females(ZW) and males (ZZ)

females (WW) and males (WY)?

Or at the very least to plot this for only the WW/WY individuals?

- Sanger sequencing: I apologize if I missed it, but I could not find how these loci were chosen for sequencing. Are they just all the loci that had a sex-linked SNP in the crosses? A few sentences in the methods and results would be very helpful even if the details are in a supplementary file.

- P5L120: "generated three families" -- I could not find information about the third family. Is this a typo? If not, how many individuals were sequenced? Did you find evidence of a ZW system there?

- P7L184: "we were able to identify at least one 100% sex-linked SNP in the Ghana east laboratory cross and the Ghana east wild population" -- This makes it sound as if the same SNP was fully sex-linked in the crosses and in the wild population, which does not seem to be the case if I understand Table 1 (i.e. no locus is classified as "Y" in both columns).

It also seems that only a single female was sanger-sequenced for the wild Ghana east population (and only 1 to 3 males for these "Y" loci), so it seems a bit optimistic to say that you found a 100% sex-linked SNP in this population?

- I found the title confusing. What is high intensity evolution, and what evidence of it do you find? What would low intensity evolution of a sex chromosome look like?

Other:

P5L114: I think quite a bit is known about when sex chromosomes become differentiated, and maybe a couple of citations would make sense here.

P5L131: "YY individuals can only be generated if"  from the rest of the sentence it sounds like they can in fact not be generated? I was a bit confused by this.

Comparing the expression of the sex-biased genes to the average expression in the sample does not seem very reliable. A better approach would be to use X. laevis gene expression as a proxy for ancestral expression to check if there has been up- or down-regulation (e.g. https://www.ncbi.nlm.nih.gov/Traces/study/?acc=SRP167133 ). But this is merely a suggestion as it is not necessarily within the scope of the present manuscript.

Similarly, it seems a shame not to have taken advantage of the SNP information in the RNA-seq data, especially given that the RAD-seq data seems to have somewhat low/inconsistent coverage.

**Have all data underlying the figures and results presented in the manuscript been provided?**

Reviewer #1: None

Reviewer #2: Yes

PLOS authors have the option to publish the peer review history of their article (what does this mean?). If published, this will include your full peer review and any attached files.

Reviewer #1: No

Reviewer #2: No

---

## [Decision Letter · Decision Letter 1]

8 Aug 2020

Dear Dr Evans,

Thank you very much for submitting your Research Article entitled 'A frog with three sex chromosomes that co-mingle together in nature: Xenopus tropicalis has a degenerate W- and a Y- that evolved from a Z-' to PLOS Genetics. Your manuscript was fully evaluated at the editorial level and by independent peer reviewers. The reviewers appreciated the attention to an important problem, but raised some substantial concerns about the current manuscript. Based on the reviews, we will not be able to accept this version of the manuscript, but we would be willing to review again a much-revised version. We cannot, of course, promise publication at that time.

If you decide to revise the manuscript for further consideration at PLOS Genetics, please aim to resubmit within the next 60 days, unless it will take extra time to address the concerns of the reviewers, in which case we would appreciate an expected resubmission date by email to plosgenetics@plos.org.

[LINK]

We are sorry that we cannot be more positive about your manuscript at this stage. Please do not hesitate to contact us if you have any concerns or questions.

Yours sincerely,

Catherine L. Peichel

Guest Editor

PLOS Genetics

Kirsten Bomblies

Section Editor: Evolution

PLOS Genetics

Comments from the Guest Editor

Both reviewers and I agree that the manuscript is improved with the new analyses. The results are quite interesting in that there is evidence that a new Y chromosome has evolved from an ancestral Z chromosome. Of course, confirming this hypothesis will take a much more detailed molecular analyses than that presented here, but this study lays the groundwork for these longer-term analyses. The main challenge is that the current manuscript is a difficult read that makes the reader work extremely hard to understand what has been found and the significance of the findings. It feels like the results have not yet been fully digested by the authors of the paper and that they can no longer “see the forest for the trees”. Reviewer 2 points out that the manuscript is still written in a historical manner and suggests some re-organisation that might help. I agree with this suggestion, as I detail below. My extensive (although not comprehensive) comments are aimed at helping the authors present their results more clearly and concisely. In addition, and as noted by Reviewer 1, I also found many small typos and grammatical errors throughout the manuscript. Not all have been indicated, and it would probably be useful for someone not associated with the study to give this manuscript a very careful read before it is resubmitted.

I am happy to re-consider another version of this manuscript, if the authors are willing to significantly revise in order to make it clearer and more accessible.

Overall organisation

1. I agree with Reviewer 2 that it would help the reader a lot to present some of the results of the analyses of the RNA-seq data of Family 3 in the first part of the paper. Although there is no RNA-seq data from the parents of this cross, the analyses of nucleotide diversity in these data is extremely illuminating for inferring the sex chromosome status of the parents of both Families 2 and 3. As such, this should be presented sooner. Essentially, I would move the text from L311-344 to the first section of the Results, after presenting the mapping results for Families 1 and 2. The results in L345-374 seem a little tangential could be shortened and/or moved to the Supplement (see more detailed comments below).

An even more radical rewrite might involve presenting the nucleotide diversity analyses from the RNA-seq analyses of Family 3 first, and then present the mapping results of Families 1 and 2. This could helpful because it would provide the probable sex chromosome status of the parents, at least of Family 2. But it is hard to know which is better without trying.

Still even more radical would be to reorder the entire results section. I would start with the population structure analyses. This would provide the reader with a better context for the populations studied here. Then, the Fst analyses could come next, as it identifies a possible sex-linked region in this broader set of populations and shows that it is the same genomic region as the previously identified W-chromosome. Then, I would follow this with the more detailed “linkage” analyses of families 1-3 in which the various sex chromosomes are identified. Then, this could be followed by the RNA-seq expression analyses, which would now be much easier to understand because we know the sex chromosome genotypes of the parents of this cross. Finally, investigation of genome-wide recombination rates could come at the end.

I leave it to the authors to determine the optimal organisation of the manuscript, but at least some changes to the organisation and clarity need to be made, as the current order made things unclear for me and for the other two reviewers.

2. The presentation of the mapping results for Families 1 and 2 is still extremely confusing and hard to follow. Here is my interpretation of the data with a suggestion for a concise summary:

Five sex-linked RRGS markers were found in the region between 8.1Mb and 13.58Mb on chromosome 7 in Family 1, while in Family 2, three sex-linked RRGS markers were found in the region between 2.7Mb and 6.54Mb on chromosome 7. However, there were no informative RRGS markers between 6.54Mb and 11Mb in Family 2, so it was not possible to assess whether RGGS markers in this region were also sex-linked. Genotyping of additional markers in Family 2 by Sanger sequencing found three sex-linked markers located between 8.1Mb and 10.26Mb, suggesting that this region is sex-linked in both Families 1 and 2. [I know it is not perfect sex-linkage, which might require a little more explanation, but not much – just that recombination occurs outside the sex-determination region!].

By contrast, informative RRGS markers between 0 and 8.1Mb were present in Family 1, but were not sex-linked in this family. [then here you can use the discussion of reasons why found in L180-190, although this could be more clearly written]. Here you could also integrate the Sanger data for this family into the discussion…

Having a separate discussion of the Sanger data is just confusing and long-winded. As suggested above, I think it would be better to integrate the Sanger and RRGS data for each family/population, with perhaps the brief paragraph on the analyses of the Sanger data in the Nigeria strain (L212-219) at the end of this section. The table with the Sanger data could probably be Supplemental.

Detailed comments:

1. L78: this sentence about the identity of master sex determining genes is not really connected to the rest of the paragraph. It could easily be deleted. If it remains, it needs to be clarified how it is related to the other information in this paragraph.

2. L82-85: this is an example of a long sentence with several spelling and grammatical errors, which could be shortened, perhaps to “Specifically, these transition periods may offer insights into whether and how characteristics of ancestral sex chromosomes (e.g. nucleotide divergence, sex-biased expression, degeneration [not degeneracy]) affect the evolution of the sex chromosome systems that follow.”

3. L141: “which can be combined in six ways for reproduction” is not correct, you are showing the possible offspring genotypes. Just say “and six possible offspring genotype combinations”. [This new Figure 1 is super helpful!!!].

4. L155: “We intentionally sampled” makes it seem like a subset of the offspring were sampled for genotyping, but you are claiming that there are equal sex ratios in this cross. Whether you genotyped all offspring in the cross, or a subset should be clarified. If a subset, you should give the total numbers so a reader can see whether there was an equal sex ratio in the cross.

5. Figure 2: Please add the Family names to the labels; i.e. Family 1 Ghana west, and Family 2 Ghana east, as you are referring to families in the text, making it challenging for the reader to connect the text and figure.

6. Figure 2 legend: As suggested by Reviewer 1, “Manhattan plot of sex linkage” is not really a clear description. Perhaps “Manhattan plot of association between genotype and sex phenotype on chromosome 7”. Please delete “the” before Family 1. Also, I could not really see the difference in the dark and light dots between the top and bottom graphs, and this explanation in the legend “Dark and light dots indicate variants with a significant or not significant association with sex” seems to refer to the darker dots which have a strong association with sex, and the lighter dots on the rest of the chromosome. So, then the phrase “respectively, after FDR correction in (top) and before FDR correction in (bottom) is very unclear. And not all three shades of blue are shown in the top legend of the figure. This needs to be clarified.

7. L221-222: Here it would be good to clarify how many individuals of each sex and population were used for the Fst analyses.

8. L232-234: The RRGS data from Family 2 only showed evidence for sex-linked markers between 0 and 6 Mb. Perhaps just say, “in the sex-linked region (between 0 and 11Mb) of chromosome 7 that was identified in the mapping families.

9. L234: by “genome-wide Fst peak” do you mean that the highest Fst value in the genome was found at this location?

10. L240: “female-linked linkage group” could be “female-linked genomic region”.

11. L241-245: this additional information about the overlap between these Sanger SNPs and the female-linked regions in the previous study does not seem necessary. You have already established that the regions of differentiation here are the same as in previous studies. If you do keep this text, please also change the “female-linked – linkage group” in line 242 to “female-linked genomic region” because it reads like it is a separate linkage group when you just mean that it is a separate region on chromosome 7.

12. L245: you conclude this paragraph by saying the sex-linked region is between 0 and 10.4Mb [I agree with Reviewer 1 that using <10.4Mb is not very clear], but you started the paragraph by saying it was between 0 and 11Mb. Please be consistent here and throughout the manuscript. Or define that the region of high Fst was between 0 and 10.4Mb. Also, I think it would be better to start this paragraph by first stating the genome-wide Fst, then the average across the sex-linked region, and then finally the peak marker. This provides better context for the reader to see that the peak Fst of 0.13 is truly a peak.

13. L255: perhaps provide the coordinates of the markers used for genotyping by Sanger sequencing.

14. L256-260: by moving the analyses of the nucleotide diversity in Family 3 earlier, you do not have to include this lengthy discussion as you would have already given the probable genotypes of this cross!

15. L264: clarify that these scaffolds were not part of chromosome assemblies

16. L266-267: this first sentence could be deleted, as well as “for example” in L268

17. L271: here you say the region of sex-linkage is less than 10.4Mb, and in L310, you say it is less than 11Mb. Another example of inconsistencies in the manuscript.

18. L311-344: as suggested, this section could be moved earlier, and also shortened.

19. L345-374: these points seem more relevant to the analyses of the sex chromosome complement of the crosses, rather than the analyses of gene expression. I think they could also be moved earlier, but also be shortened and/or be supplemental.

20. L375-398: I really like these new analyses! It would seem to follow most naturally after the analyses of the male vs female RNA-seq analyses (especially if you follow the suggestion to move the analyses of nucleotide divergence earlier in the manuscript). In these paragraphs, please also change “degeneracy” to “degeneration of”.

21. Figure 6: As in Figure 2, please label the graphs as Family 1 Ghana west, and Family 2 Ghana east.

22. L407-408 and L412: “the size in base pairs of the genomic region to which the linkage map corresponds” is so awkward and wordy! Perhaps change to the “physical size in base pairs of the chromosome assembly”. Also, here you say “females maps” and “males maps” instead of female and male maps.

23. L428: “subdivision is within” to “subdivisions are found within”

24. L443: “two localities Ghana” should be “two localities in Ghana”

25. L446-447: how are your findings consistent with the origin of the Y in Nigeria? You found evidence for a Y in Nigeria, but it’s not clear how that is evidence that it originated there.

26. L448: It might be helpful to say where the Ivory Coast populations are located, relative to those shown in Figure 7.

27. L463-465: this sentence is hard to parse. Also, just because you found these chromosomes in your crosses, doesn’t mean they are common. You might have just gotten lucky. Thus, I don’t think you can say anything about whether they are on the brink of extinction.

28. L499-501: less than “one hundred and fiftieth”: can you just give the percentage; i.e. less than 1% of the genome? And, it doesn’t seem necessary to give two different ways to characterize the abundance of sex-biased expression in this region of the genome. This is quite clear already.

29. L503-506: there is an interesting paper by Jun Kitano’s group (Kitano et al. 2020 Journal of Evolutionary Biology doi: 10.1111/jeb.13662), suggesting that sex-biased expression at earlier stages of development is more likely to be due to sex-linkage, while sex-biased expression at later stages of development is more likely to be due to hormonal mechanisms. These results are also consistent with that hypothesis.

30. L517-518: I don’t quite understand the point here.

31. L519-521: this sentence doesn’t seem necessary, but the faster-Z should probably be briefly explained in the next sentence.

32. L541: “each one generation in length” could be deleted.

33. L570: “evidence W-chromosome degeneracy” could be “evidence for W-chromosome degeneration”.

34. L589-591: I think this information comes later, but it would be good to provide the numbers of wild-caught males and female samples in each population used for RRGS.

35. L613-614: “or for the offspring” seems incomplete.

36. L619: “east” should be “Family 2” for clarity.

37. L631-633: in the absence of providing any details about these markers, there should at least be a reference the Table with this marker information.

38. L650: “likelihood of their being a SNP” should be “likelihood of there being a SNP”

39. L665-666: it would be good to provide the genomic location of these markers used to genotype the tadpoles.

Reviewer's Responses to Questions

**Comments to the Authors:**

Reviewer #1: The manuscript is improved by the extra analysis, but it remains poorly written and largely descriptive of a further study of a case of within-species variation in the heterogametic sex. Much of the text is long-winded and difficult to understand, obscuring the main points. It is valuable to get genetic data, and this appears to establish that there is no physically extensive fully sex-linked region in the species studied, consistent with its apparently homomorphic sex chromosomes, and with the fact (already known) that it has a sex chromosome polymorphism with Z, W and a more recently derived Y chromosomes, similar to the polymorphisms in some other frogs.

The study may indeed have identified, for the first time in natural X. tropicalis population samples, the Y-chromosome previously found in captive individuals. The paragraph starting in line 442 is very long, and not easy to follow. The information needs to be digested and presented more clearly after the authors have decided what they need to tell readers. I hoped for an advance in understanding, or at least a clear discussion, of whether the results support one model for turnover events or another, but it is not clear to me what these results tell us about turnovers.

The most interesting other result is that there are unexpectedly many transcripts with significantly male-biased or (not “and” as written in line 278) male-specific expression on chromosome 7. I do not have the necessary expertise to evaluate this, but obviously it is important to use reference genome from an appropriate sex in order to avoid a bias due to potential absence of some sequences in some fully sex-linked regions. The paper by Wei and D. Bachtrog, 2019 is now cited, but this problem, and the reference assemblies used, should surely be mentioned in the main text, not just a Supplementary file.

The Discussion (from line 495) deals with this finding as a “Sign of Age in Cytologically Indistinguishable Sex Chromosomes”, but I think the authors probably mean “sign of adaptation” or maybe “sign of genetic degeneration of the W”. Again the text could be clearer after shortening to make the essential point(s) clear. In the present version, the evidence for the claimed genetic degeneration of the W chromosome is not compelling, and needs to be much clearer.

The following text about these expression results is again very hard to understand, but may mean the following: We therefore examined the genotypes in sex-linked expressed transcripts of each individual offspring of Family 3 using the RNAseq data, in order to detect transcripts expressed from only one of the individual’s sex chromosomes (based on observing no heterozygous variants), and those co-expressed by both gametologous alleles in heterozygous genotypes [17]. I leave the rest of this section for the authors to revise and clarify. It is very confusing to write about “levels of polymorphism” when genotypes are meant. Possibly a table showing the possibilities (including a degenerated W-chromosome), and the expectations in the different situations and the different parental genotypes, would be helpful. Ideally the results would be digested and shown as a comparison with those expectations, so that the text can guide the reader to understand the evidence for the

results about the sex-specificity of rates and locations of recombination in X. tropicalis are potentially helpful for understanding how and/or why a turnover happened, though I still found the conclusions rather weakly supported. The results are consistent with the conclusion, but not compelling. The discussion of what we can learn from this is improved, but still appears not to be related to the claimed genetic degeneration of the W chromosome.

The section about population structure needs an introduction to give an idea of why these data are needed. This could maybe be merged with the start of the Discussion.

MINOR COMMENTS

The writing still needs considerable revision, as it is often very long-winded, making it difficult to understand the meaning.

Throughout, the manner of specifying regions on a chromosome like this example “<10.4Mb” should be changed to be clear to readers. There should also be no hyphens in the phrase Y (or other) chromosome. The authors should read the text carefully for errors (I have not listed all of them here), and make sure that the word “that” is not missing in places where it is needed in written English.

SHORTENINGS AND EDITINGS

Line 67: Shorten to: A sex chromosome turnover is called "homologous" when a new variant that assumes the sex determination role arises on an ancestral sex chromosome [1], and "non-homologous" if it establishes a on a different chromosome pair from the ancestral sex chromosomes.

Line 74 Turnovers 9in the plural)

Line 75:. Non-homologous XY to XY turnovers may be favoured by natural selection if the ancestral Y-chromosome has a high load of deleterious mutations due to genetic degeneration [21,22]. However, Y-linked deleterious mutations may disfavour XY to WZ transitions because they result in the appearance of homozygotes for the ancestral Y-chromosome[16].

Surely a new paragraph is needed before “So far, only a handful of master sex determining genes are known”.

Line 82 Specifically, these transitions [should be singular] periods may offer insights ….

Line 95: Although it is technically no longer a Z-chromosome after the Y-chromosome appeared, we

use this term as a placeholder to refer to the extant non-male-specific sex-chromosome that descended from 97 the ancestral Z-chromosome, following [28]” can be shortened to “We use the term ‘Z-chromosome’, following [28]’, to refer to the extant non-male-specific sex-chromosome that presumably evolved from the ancestral Z-chromosome still present in related species”. Is this the correct meaning? How is it known that this Z is the ancestral one, and that the Y is derived? If the evidence is in this paper, why not make that clear here?

Line 102: The genomic location of the female-associated region of the W-chromosome in a laboratory strain was narrowed down [BY WHAT APPROACH?] to an interval between positions 0–3.9 megabases (Mb) on chromosome 7 in genome assembly 9.1 (v9) [30]. However, this region did not show complete linkage to the female phenotype in our study (below), and it was proposed that this could be due to ancestral admixture with an individual carrying a Y-chromosome [30]. The male determining factor of the X. tropicalis Y-chromosome of is thought to be in a similar location to that of the female-determining factor [28], but has nor been precisely located. Within the genus Xenopus, the most recent common ancestor of subgenus Silurana, which includes X. tropicalis, probably had heterogametic females [26], implying that the X. tropicalis Y chromosome is derived from an ancestral W or Z chromosome. Mitochondrial genomes of species in subgenus Silurana diverged about 25 million years ago [33], implying that the Y-chromosome of X. tropicalis is younger than that. This

113 variation raises the possibility that X. tropicalis is currently undergoing a homologous sex chromosome turnover. HERE IT IS NOT CLEAR WHAT ‘variation’ IS BEING MENTIONED.

Line 122: We also evaluate whether and how recombination differs between the sexes of X. tropicalis – including in the sex-linked region and across the genome. The goals of this study are to ….

Line 131 : a comma is needed before “and their offspring”

Line 135: We also studied sex linkage in different portions of the sex-linked region, using Sanger sequencing of selected amplicons.

Line 140: There are two possible sex chromosome genotypes in females (WZ, WW) and three in males (ZZ, ZY, WY), and six possible parent genotype combinations (Fig. 1).

Line 170: None of the sex-linked SNPs from Families 1 and 2 were present in the sequences of the other family, presumably reflecting variable presence of SbfI restriction sites. In each of the Ghana families, some SNPs in the region that was sex linked in one family displayed genotypes suggesting independent segregation in the other (Fig. 2). One possible explanation is that our sex-linked markers are partially sex-linked. In Ghana East Family 2, we lacked markers between 6-11Mb on chromosome 7, and therefore our the RRGS data cannot delimit the fully sex-linked region In this family. THE QUESTION IS WHETHER THESE MARKERS ARE PARTIALLY SEX-LINKED OR SEGREGATE INDEPENDENTLY OF THE SEX-DETERMINING LOCUS.

Line 179: The writing needs correcting here (the current version reads rather raw and undigested) — “On chromosome 7 <6Mb, we had variable markers that overlapped in Family 1 and 2, and these markers were male-linked in Family 2 but not Family 1 (Fig. 2). Analyses discussed below allowed us to infer….This scenario would explain why there were sex-linked sites on the end of chromosome 7 in Family 2 but not Family 1; it is also consistent with evidence presented below for a lack of recombination in the sex-linked region during spermatogenesis of male BJE4362, the father of families 2 and 3, and with an origin of the Y chromosome from an ancestral Z chromosome

Figure 2 legend: I don’t think one can just sat “Manhattan plot of sex linkage for chromosome….”

Line 193: Sanger sequencing identified one SNP that co-segregates in our small Family 2 with the sex-determining locus, and two almost completely co-segregating SNPs. With a very small sample size we identified three completely co-segregating SNPs in the Ghana east

wild population, but none in either Family 1 or the Ghana west wild population. The lack of sex-specific SNPs (even with our small sample) in Family 1 is consistent with the hypothesis proposed above that the sex chromosome genotype of the father may have the ZY genotype

(though other explanations are possible).

Line 216: However, without invoking Y linkage, the chance of observing a heterozygous genotype in 17 of 17 males and none of 18 females is very low (_ 7 _ 106).

This sentence is hard to understand, and so is its relevance (and the tense is wrong): “The sex-linked SNPs in the Nigeria strain was different from a nearby sex-linked SNP in three wild Ghana east males.”

Line 220: The X. tropicalis Sex Chromosomes Have ONLY a Small Differentiated Region

Line 223: This passage is particularly badly written and can be much shorter and clearer. As demonstrated above, some of these individuals have a Y chromosome, and , because a W chromosome is required for female development, they must all have a W; some individuals of either sex may have a Z chromosome along with their Y or W. Differences in allele frequencies are expected to affect FST between females and males, and nucleotide divergence between the Y, Z and W chromosomes leads to different frequencies in the two sexes, and can reveal fully sex-linked regions, with higher differentiation than elsewhere in the same chromosome.

We indeed observed higher FST values in the region where sex-linkage was detected in our analyses above using RRGS data from Family 2 (the chromosome 7 region before 11Mb), with an FST value of 0.13 at position 9,940,000 in the v10 assembly of this chromosome, and values >0.09 between positions 9,775,600 and 9,999,600 (1,615,479 to 1,454,645 – 1,664,477 in assembly v9). The genome-wide 95% CI is 0.002 – 0.038. [THIS SHOULD BE SPECIFIED HERE, NOT LATER]. Are the high FST values due to low diversity within the population of X (or other relevant) chromosomes?

Line 239 : should be changed to “found previously [30] (Suppl.Fig. S3).”

Line 266: This sentence is not clear (and can probably be omitted). For neutral variants in an autosomal locus, genetic drift of the transcriptome is not expected to produce a skew in sex-biased expression.

Line 304” another problem with tenses — “it does suggest that there is no reason to expect that transcripts in this genomic region was somehow predisposed to have male-biased expression”

Line 307: There is no need to keep repeating the information that high densities of male- or female-biased expression are rare on the autosomes. This is repeated 3 times in quick succession.

Line 311: section on How might sex-linkage lead to a skew towards male-biased expression of transcripts encoded by sex linked genes? Again, the writing is vague and unclear. I think the first sentence (One possible explanation is that expression of some alleles in the sex-linked region of the chromosome decreased or was [wrong tense again] lost due to recombination suppression) is referring to genetic degeneration, but it is hard to be sure if this is the meaning. Maybe it is supposed to mean “One possibility is that expression of some alleles in the sex-linked region of the chromosome became decreased below the level of alleles on an ancestral non-degenerated Z-chromosome, and thus alleles that are now Y-linked (and were derived from a Z) retain the Z’s high ancestral expression level”.

Line 432: “interceded” is the wrong word here.

Reviewer #2: I think that the current version of the paper is much stronger - the analyses and interpretations are sound, and the results still very exciting. The authors did a great job at addressing my previous comments (although I would specify "but only a single female and 1 to 3 males were sequenced" rather than saying "with a very small sample size").

My only remaining comment is that I still sometimes found the manuscript a bit hard to follow (but figure 1 is very helpful!).

Some of that seems to come from the fact that the structure of the paper reflects its history. I think it would make more sense to have the inference of the genotypes of the three families in the first section (including the RNA), and then going into the gene expression knowing those genotypes, instead of going back to them later.

In general the text seems (to me) a bit longer than necessary. For instance, the Sanger sequencing part, from which not much was concluded, could maybe be shorter or even moved to the supplementary material.

**Have all data underlying the figures and results presented in the manuscript been provided?**

Reviewer #1: Yes

Reviewer #2: None

PLOS authors have the option to publish the peer review history of their article (what does this mean?). If published, this will include your full peer review and any attached files.

Reviewer #1: No

Reviewer #2: No

---

## [Editor Report · Decision Letter 2]

16 Sep 2020

Dear Dr Evans,

We are pleased to inform you that your manuscript entitled "A frog with three sex chromosomes that co-mingle together in nature: Xenopus tropicalis has a degenerate W and a Y that evolved from a Z chromosome" has been editorially accepted for publication in PLOS Genetics. Congratulations!

Yours sincerely,

Catherine L. Peichel

Guest Editor

PLOS Genetics

Kirsten Bomblies

Section Editor: Evolution

PLOS Genetics

Comments from the reviewers (if applicable):

I would like to thank the authors for their careful revisions. The manuscript has been greatly improved! It is a complex set of results, but the main results and conclusions are far more clearly presented now. This will be a nice contribution to the literature on sex chromosome turnovers.

I still have a very few minor comments, but I think these can be dealt with before the final files are uploaded rather than as a revision.

P4, L90: Can delete the full spelling out of Xenopus tropicalis as it appears in the line before.

P4, L97: YZ should be ZY to be consistent with the rest of the manuscript.

P5, L149-154: This would be much better to state in the introductory paragraph to the Results, where you nicely provide an overview of what you did. Here, it breaks the flow.

P5, L151: Here you reference Fig. S3 without referencing Fig. S1 or S2. I checked and the order of calling out the supplementary figures and tables is jumbled (probably due to the reorganization). Please renumber and reorder the supplementary figures and tables so that they are numbered in the order that they appear in the manuscript.

P7, L158: delete “genome wide”

P7, L197: Here you define the sex-linked region with a marker at 10.26Mb, but later you say consistently say that the sex-linked region is found at less than 10.4Mb. Perhaps this needs to be clarified somewhere.

P7, L218-219: perhaps change to “homozygous genotypes in one son and heterozygous in four other sons”

P7, L223: perhaps delete “at the NXR” because it is not clear what this is, and these details should be in the methods.

P7, L229: add “P” before “<7x10-6”

P9, L237-238: maybe change “gametologous alleles” to “both alleles”

P12, L301: maybe change “gametologous loci” to “gametologs”

The version of Figure 3 in my PDF is missing the grey band and the x and y-axis lines.

**Data Deposition**

http://datadryad.org/submit?journalID=pgenetics&manu=PGENETICS-D-20-00636R2

**Press Queries**

---

## [Editor Report · Acceptance letter]

15 Oct 2020

PGENETICS-D-20-00636R2 

A frog with three sex chromosomes that co-mingle together in nature: Xenopus tropicalis has a degenerate W and a Y that evolved from a Z chromosome 

Dear Dr Evans, 

We are pleased to inform you that your manuscript entitled "A frog with three sex chromosomes that co-mingle together in nature: Xenopus tropicalis has a degenerate W and a Y that evolved from a Z chromosome" has been formally accepted for publication in PLOS Genetics! Your manuscript is now with our production department and you will be notified of the publication date in due course.

With kind regards,

Matt Lyles

PLOS Genetics

On behalf of:
